# Beyond Concept Bottleneck Models: How to Make Black Boxes Intervenable?

## Abstract

Recently, interpretable machine learning has re-explored concept bottleneck models (CBM), comprising step-by-step prediction of the high-level concepts from the raw features and the target variable from the predicted concepts. A compelling advantage of this model class is the user's ability to intervene on the predicted concept values, consequently affecting the model's downstream output. In this work, we introduce a method to perform such concept-based interventions on already-trained neural networks, which are not interpretable by design. Furthermore, we formalise the model's *intervenability* as a measure of the effectiveness of concept-based interventions and leverage this definition to fine-tune black-box models. Empirically, we explore the intervenability of black-box classifiers on synthetic tabular and natural image benchmarks. We demonstrate that fine-tuning improves intervention effectiveness and often yields better-calibrated predictions. To showcase the practical utility of the proposed techniques, we apply them to deep chest X-ray classifiers and show that fine-tuned black boxes can be as intervenable and more performant than CBMs.

## 1 Introduction

Interpretable and explainable machine learning (Doshi-Velez & Kim, 2017; Molnar, 2022) have seen a renewed interest in concept-based predictive models and approaches to post hoc explanation, such as concept bottlenecks (Lampert et al., 2009; Kumar et al., 2009; Koh et al., 2020), contextual semantic interpretable bottlenecks (Marcos et al., 2020), concept whitening layers (Chen et al., 2020), and concept activation vectors (Kim et al., 2018). Moving beyond interpretations defined in the high-dimensional and unwieldy input space, these techniques relate the model's inputs and outputs via additional high-level human-understandable attributes, also referred to as concepts. Typically, neural network models are supervised to predict these attributes in a dedicated bottleneck layer, or post hoc explanations are derived to measure the model's sensitivity to a set of concept variables.

This work focuses specifically on the concept bottleneck models, as revisited by Koh et al. (2020). In brief, a CBM $f_{\boldsymbol{\theta}}$, parameterised by $\boldsymbol{\theta}$, is given by $f_{\boldsymbol{\theta}}(\boldsymbol{x}) = g_{\boldsymbol{\psi}}(h_{\boldsymbol{\phi}}(\boldsymbol{x}))$, where $\boldsymbol{x} \in \mathcal{X}$ and $y \in \mathcal{Y}$ are covariates and targets, respectively, $h_{\boldsymbol{\phi}} : \mathcal{X} \to \mathcal{C}$ maps inputs to predicted concepts, *i.e.* $\hat{\boldsymbol{c}} = h_{\boldsymbol{\phi}}(\boldsymbol{x})$, and $g_{\boldsymbol{\psi}} : \mathcal{C} \to \mathcal{Y}$ predicts the target based on $\hat{\boldsymbol{c}}$, *i.e.* $\hat{y} = g_{\boldsymbol{\psi}}(\hat{\boldsymbol{c}})$. CBMs are trained on labelled data points $(\boldsymbol{x}, \boldsymbol{c}, y)$ annotated by concepts $\boldsymbol{c} \in \mathcal{C}$ and are supervised by the concept and target prediction losses. Note that above, the output of $h_{\boldsymbol{\phi}}$ forms a *concept bottleneck layer*, and thus, the model's final output depends on the covariates $\boldsymbol{x}$ solely through the predicted concept values $\hat{\boldsymbol{c}}$. At inference time, a human user may interact with the CBM by editing the predicted concept values and, as a result, affecting the downstream target prediction. For example, if the user chooses to replace $\hat{\boldsymbol{c}}$ with another $\boldsymbol{c}' \in \mathcal{C}$, the final prediction is given by $\hat{y}' = g_{\boldsymbol{\psi}}(\boldsymbol{c}')$. This act of model editing is known as an *intervention*. User's ability to intervene is a compelling advantage of CBMs over other interpretable model classes, in that the former allows for human-model interaction.

An apparent limitation of the CBMs is that the knowledge of the concepts and annotated data are required at model development. Recent research efforts have been directed at mitigating these limitations by converting pretrained models into CBMs post hoc (Yuksekgonul et al., 2023) and discovering concept sets automatically in a label-free manner using GPT-3 and CLIP models (Oikarinen et al., 2023). However, these works either have not comprehensively investigated the effectiveness of interventions in this setting or have mainly concentrated on global model editing rather than

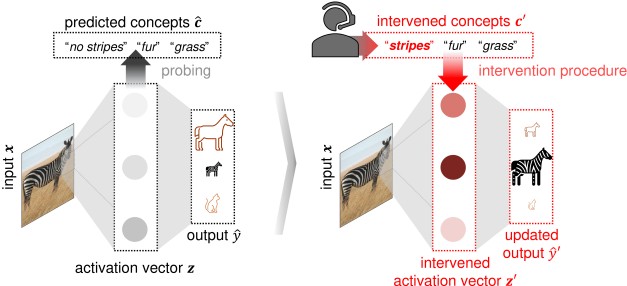

Figure 1: Schematic summary of concept-based instance-specific intervention on a black-box neural network. This work introduces an intervention procedure that, given concept values $c'$, for an input $x$, edits the network's activation vector $z$ at an intermediate layer, replacing it with $z'$ coherent with the given concepts. The intervention results in an updated prediction $\hat{y}'$.

the influence on individual data point predictions. Complementary to the post hoc and label-free CBMs, we focus on interventions and explore two related research questions: (i) *Given post hoc a concept set and dataset with concept labels, how can we perform instance-specific interventions directly on a trained black-box model?* (ii) *How can we fine-tune the black-box model to improve the effectiveness of interventions performed on it?* Herein, instance-specific interventions refer to the interventions performed locally, *i.e.* individually for each data point. Figure 1 schematically summarises the principle behind concept-based interventions on black-box neural network models.

**Contributions** This work contributes to the line of research on concept bottleneck models and concept-based explanations in several ways. (1) We devise a simple procedure (Figure 1) that, given a set of concepts and a labelled dataset, allows performing concept-based instance-specific interventions on an already trained black-box neural network by editing its activations at an intermediate layer. Notably, during training, concept labels are not required and the network's architecture does not need to be adjusted. (2) We formalise *intervenability* as a measure of the effectiveness of the interventions performed on the model and introduce a novel fine-tuning procedure for black-box neural networks that utilises intervenability as the loss. This fine-tuning strategy is designed to improve the effectiveness of concept-based interventions while preserving the original model's architecture and learnt representations. (3) We evaluate the proposed procedures alongside several common-sense baseline techniques on the synthetic tabular, natural image, and medical imaging data. We demonstrate that in practice, for some classification problems, we can improve the predictive performance of already trained black-box models via concept-based interventions. Moreover, the effectiveness of interventions improves considerably when explicitly fine-tuning for intervenability.

## 2 RELATED WORK

The use of high-level attributes in predictive models has been well-explored in computer vision (Lampert et al., 2009; Kumar et al., 2009). Recent efforts have focused on explicitly incorporating concepts in neural networks (Koh et al., 2020; Marcos et al., 2020), producing high-level post hoc explanations by quantifying the network's sensitivity to the attributes (Kim et al., 2018), probing (Alain & Bengio, 2016; Belinkov, 2022) and de-correlating and aligning the network's latent space with concept variables (Chen et al., 2020). To alleviate the assumption of being given interpretable concepts, some works have explored concept discovery prior to post hoc explanation (Ghorbani et al., 2019; Yeh et al., 2020). Another relevant line of work investigated concept-based counterfactual explanations (CCE) (Abid et al., 2022; Kim et al., 2023b).

Concept bottleneck models (Koh et al., 2020) have sparked a renewed interest in concept-based classification methods. Many related works have described the inherent limitations of this model class and attempted to address them. For example, Margeloiu et al. (2021) observe that CBMs may not always learn meaningful relationships between the concept and input spaces. Similarly, Mahinpei et al. (2021) identify the issue of information leakage in concept predictions. Solutions to this challenge include generative approaches (Marconato et al., 2022) and learning residual relationships between the features and labels that are not captured by the given concept set (Havasi et al., 2022;

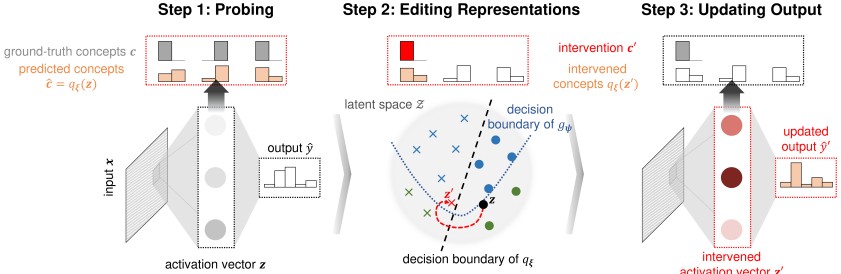

Figure 2: Three steps of the intervention procedure. (i) A probe $q_{\boldsymbol{\xi}}$ is trained to predict the concepts $\boldsymbol{c}$ from the activation vector $\boldsymbol{z}$. (ii) The representations are edited according to Equation 1. (iii) The final prediction is updated to $\hat{y}'$ based on the edited representations $\boldsymbol{z}'$.

Sawada & Nakamura, 2022; Marcinkevičs et al., 2023). Another line of research has investigated modelling uncertainty and probabilistic extensions of the CBMs (Collins et al., 2023; Kim et al., 2023a). Most related to the current work are the techniques for converting already trained black-box neural networks into CBMs post hoc (Yuksekgonul et al., 2023; Oikarinen et al., 2023) by keeping the network's backbone and projecting its activations into the concept bottleneck layer.

As mentioned, CBMs allow for concept-based instance-specific interventions. Several follow-up works have studied interventions in further detail. Chauhan et al. (2023) and Sheth et al. (2022) introduce adaptive intervention policies to further improve the predictive performance of the CBMs at the test time. In a similar vein, Steinmann et al. (2023) propose learning to detect mistakes in the predicted concepts and, thus, learning intervention strategies. Shin et al. (2023) empirically investigate different intervention procedures across various settings.

## 3 METHODS

In this section, we define a measure for the effectiveness of interventions and present techniques for performing concept-based interventions on black-box neural networks and fine-tuning black boxes to improve the effectiveness of such interventions. Some additional remarks beyond the scope of the main text are included in Appendix B. In the remainder of this paper, we will adhere to the following notation. Let $\boldsymbol{x} \in \mathcal{X}$, $y \in \mathcal{Y}$, and $\boldsymbol{c} \in \mathcal{C}$ be the covariates, targets, and concepts. Consider a black-box neural network $f_{\boldsymbol{\theta}} : \mathcal{X} \rightarrow \mathcal{Y}$ parameterised by $\boldsymbol{\theta}$ and a slice $\langle g_{\boldsymbol{\psi}}, h_{\boldsymbol{\phi}} \rangle$ (Leino et al., 2018), defining a layer, s.t. $f_{\boldsymbol{\theta}}(\boldsymbol{x}) = g_{\boldsymbol{\psi}}(h_{\boldsymbol{\phi}}(\boldsymbol{x}))$. We will assume that the black box has been trained end-to-end on the labelled data $\{(\boldsymbol{x}_i, y_i)\}_i$. When applicable, we will use a similar notation for CBMs, as outlined in Section 1. Lastly, for the techniques introduced below, we will assume being given a labelled and annotated validation set $\{(\boldsymbol{x}_i, \boldsymbol{c}_i, y_i)\}_i$.

### 3.1 INTERVENING ON BLACK-BOX MODELS

Given a black-box model $f_{\boldsymbol{\theta}}$ and a data point $(\boldsymbol{x}, y)$, a human user might desire to influence the prediction $\hat{y} = f_{\boldsymbol{\theta}}(\boldsymbol{x})$ made by the model via high-level and understandable concept values $\boldsymbol{c}'$, *e.g.* think of a doctor trying to interact with a chest X-ray classifier ($f_{\boldsymbol{\theta}}$) by annotating their findings ($\boldsymbol{c}'$) in a radiograph ($\boldsymbol{x}$). To facilitate such interactions, we propose a simple recipe for concept-based instance-specific interventions (detailed in Figure 2) that can be applied to any black-box neural network model. Intuitively, using the given validation data and concept values, our procedure edits the network's representations $\boldsymbol{z} = h_{\boldsymbol{\phi}}(\boldsymbol{x})$, where $\boldsymbol{z} \in \mathcal{Z}$, to align more closely with $\boldsymbol{c}'$ and, thus, affects the downstream prediction. Below, we explain this procedure step-by-step. Pseudocode implementation can be found as part of Algorithm A.1 in Appendix A.

**Step 1: Probing** To align the network's activation vectors with concepts, the preliminary step is to train a probing function (Alain & Bengio, 2016; Belinkov, 2022), or a probe for short, mapping the intermediate representations to concepts. Namely, using the given annotated validation data $\{(\boldsymbol{x}_i, \boldsymbol{c}_i, y_i)\}_i$, we train a probe $q_{\boldsymbol{\xi}}$ to predict the concepts $\boldsymbol{c}_i$ from the representations $\boldsymbol{z}_i = h_{\boldsymbol{\phi}}(\boldsymbol{x}_i)$: $\min_{\boldsymbol{\xi}} \sum_i \mathcal{L}^{\boldsymbol{c}}(q_{\boldsymbol{\xi}}(\boldsymbol{z}_i), \boldsymbol{c}_i)$, where $\mathcal{L}^{\boldsymbol{c}}$ is the concept prediction loss. Note that, herein, an essential

design choice explored in our experiments is the (non)linearity of the probe. Consequently, the probing function can be used to interpret the activations in the intermediate layer and edit them.

**Step 2: Editing Representations**    Recall that we are given a data point $(\boldsymbol{x}, y)$ and concept values $\boldsymbol{c}'$ for which an intervention needs to be performed. Note that this $\boldsymbol{c}' \in \mathcal{C}$ could correspond to the ground-truth concept values or reflect the beliefs of the human subject intervening on the model. Intuitively, we seek an activation vector $\boldsymbol{z}'$, which is similar to $\boldsymbol{z} = h_{\boldsymbol{\phi}}(\boldsymbol{x})$ and consistent with $\boldsymbol{c}'$ according to the previously learnt probing function $q_{\boldsymbol{\xi}}$: $\arg\min_{\boldsymbol{z}'} d(\boldsymbol{z}, \boldsymbol{z}')$, s.t. $q_{\boldsymbol{\xi}}(\boldsymbol{z}') = \boldsymbol{c}'$, where $d$ is an appropriate distance function applied to the activation vectors from the intermediate layer. Throughout main experiments (Section 4), we utilise the Euclidean metric, which is frequently applied to neural network representations, *e.g.* see works by Moradi Fard et al. (2020) and Jia et al. (2021). In Appendix E.5, we additionally explore the cosine distance. Instead of the constrained problem above, we resort to minimising a relaxed objective:

$$\arg\min_{\boldsymbol{z}'} \ \lambda \mathcal{L}^{\boldsymbol{c}}\left(q_{\boldsymbol{\xi}}(\boldsymbol{z}'), \boldsymbol{c}'\right) + d(\boldsymbol{z}, \boldsymbol{z}'), \tag{1}$$

where, similarly to the counterfactual explanation (Wachter et al., 2017; Mothilal et al., 2020), hyperparameter $\lambda > 0$ controls the tradeoff between the intervention's validity, *i.e.* the "consistency" of $\boldsymbol{z}'$ with the given concept values $\boldsymbol{c}'$ according to the probe, and proximity to the original activation vector $\boldsymbol{z}$. In practice, we optimise $\boldsymbol{z}'$ for batched interventions using Adam (Kingma & Ba, 2015).

**Step 3: Updating Output**    The edited representation $\boldsymbol{z}'$ can be consequently fed into $g_{\boldsymbol{\psi}}$ to compute the updated output $\hat{y}' = g_{\boldsymbol{\psi}}(\boldsymbol{z}')$, which could be then returned and displayed to the human subject. For example, if $\boldsymbol{c}'$ are the ground-truth concept values, we would ideally expect a decrease in the prediction error for the given data point $(\boldsymbol{x}, y)$.

## 3.2 What is Intervenability?

Concept bottlenecks (Koh et al., 2020) and their extensions are often evaluated empirically by plotting test-set performance or error attained after intervening on concept subsets of varying sizes. Ideally, the model's test-set performance should improve when given more ground-truth attribute values. Below, we formalise this notion of intervention effectiveness, referred to as *intervenability*, for the concept bottleneck and black-box models.

Following the notation from Section 1, for a trained CBM $f_{\boldsymbol{\theta}}(\boldsymbol{x}) = g_{\boldsymbol{\psi}}(h_{\boldsymbol{\phi}}(\boldsymbol{x})) = g_{\boldsymbol{\psi}}(\hat{\boldsymbol{c}})$, we define the intervenability as follows:

$$\mathbb{E}_{(\boldsymbol{x},\boldsymbol{c},y)\sim\mathcal{D}}\left[\mathbb{E}_{\boldsymbol{c}'\sim\pi}\left[\mathcal{L}^y\Big(\underbrace{f_{\boldsymbol{\theta}}(\boldsymbol{x})}_{\hat{y}=g_{\boldsymbol{\psi}}(\hat{\boldsymbol{c}})}, y\Big) - \mathcal{L}^y\Big(\underbrace{g_{\boldsymbol{\psi}}(\boldsymbol{c}')}_{\hat{y}'}, y\Big)\right]\right], \tag{2}$$

where $\mathcal{D}$ is the joint distribution over the covariates, concepts, and targets, $\mathcal{L}^y$ is the target prediction loss, *e.g.* the mean squared error (MSE) or cross-entropy (CE), and $\pi$ denotes a distribution over edited concept values $\boldsymbol{c}'$. Observe that Equation 2 generalises the standard evaluation strategy of intervening on a random concept subset and setting it to the ground-truth values, as proposed in the original work by Koh et al. (2020). Here, the effectiveness of interventions is quantified by the gap between the regular prediction loss and the loss attained after the intervention: the larger the gap between these values, the stronger the effect interventions have. The intervenability measure is related to permutation-based variable importance and model reliance (Fisher et al., 2019). We provide a detailed discussion of this relationship in Appendix B.

Note that the definition in Equation 2 can also accommodate more sophisticated intervention strategies, for example, similar to those studied by Shin et al. (2023) and Sheth et al. (2022). An intervention strategy can be specified via the distribution $\pi$, which can be conditioned on $\boldsymbol{x}$, $\hat{\boldsymbol{c}}$, $\boldsymbol{c}$, $\hat{y}$, or even $y$: $\pi(\boldsymbol{c}'|\boldsymbol{x}, \hat{\boldsymbol{c}}, \boldsymbol{c}, \hat{y}, y)$. The set of conditioning variables may vary depending on the specific application scenario. For brevity, we will use $\pi$ as a shorthand notation for this distribution. Lastly, notice that, in practice, when performing human- or application-grounded evaluation (Doshi-Velez & Kim, 2017), sampling from $\pi$ may be replaced with the interventions by a human. Algorithms D.1 and D.2 provide concrete examples of the strategies utilised in our experiments.

Leveraging the intervention procedure described in Section 3.1, analogous to Equation 2, the intervenability for a black-box neural network $f_{\boldsymbol{\theta}}$ at the intermediate layer given by $\langle g_{\boldsymbol{\psi}}, h_{\boldsymbol{\phi}} \rangle$ is

$$
\begin{aligned}
&\mathbb{E}_{(\boldsymbol{x},\boldsymbol{c},y)\sim\mathcal{D},\,\boldsymbol{c}'\sim\pi}\left[\mathcal{L}^y\left(f_{\boldsymbol{\theta}}\left(\boldsymbol{x}\right),y\right)-\mathcal{L}^y\left(g_{\boldsymbol{\psi}}\left(\boldsymbol{z}'\right),y\right)\right],\\
&\text{where } \boldsymbol{z}'\in\arg\min_{\tilde{\boldsymbol{z}}}\lambda\mathcal{L}^{\boldsymbol{c}}\left(q_{\boldsymbol{\xi}}\left(\tilde{\boldsymbol{z}}\right),\boldsymbol{c}'\right)+d\left(\boldsymbol{z},\tilde{\boldsymbol{z}}\right).
\end{aligned}
\tag{3}
$$

Recall that $q_{\boldsymbol{\xi}}$ is the probe trained to predict $\boldsymbol{c}$ based on the activations $h_{\boldsymbol{\phi}}\left(\boldsymbol{x}\right)$ (step 1, Section 3.1). Furthermore, in the first line of Equation 3, edited representations $\boldsymbol{z}'$ are a function of $\boldsymbol{c}'$, as defined by the second line, which corresponds to step 2 of the intervention procedure (Equation 1).

## 3.3 FINE-TUNING FOR INTERVENABILITY

Since the intervenability measure defined in Equation 3 is differentiable, a neural network can be fine-tuned by explicitly maximising it using, for example, mini-batch gradient descent. We expect fine-tuning for intervenability to reinforce the model's reliance on the high-level attributes and have a regularising effect. In this section, we provide a detailed description of the fine-tuning procedure (Algorithm A.1, Appendix A), and, afterwards, we demonstrate its practical utility empirically.

To fine-tune an already trained black-box model $f_{\boldsymbol{\theta}}$, we combine the target prediction loss with the weighted intervenability term, which amounts to the following optimisation problem:

$$
\begin{aligned}
&\min_{\boldsymbol{\phi},\boldsymbol{\psi},\boldsymbol{z}'}\mathbb{E}_{(\boldsymbol{x},\boldsymbol{c},y)\sim\mathcal{D},\,\boldsymbol{c}'\sim\pi}\left[\left(1-\beta\right)\mathcal{L}^y\Big(g_{\boldsymbol{\psi}}\left(h_{\boldsymbol{\phi}}\left(\boldsymbol{x}\right)\right),y\Big)+\beta\mathcal{L}^y\Big(g_{\boldsymbol{\psi}}\left(\boldsymbol{z}'\right),y\Big)\right],\\
&\text{s.t. } \boldsymbol{z}'\in\arg\min_{\tilde{\boldsymbol{z}}}\lambda\mathcal{L}^{\boldsymbol{c}}\left(q_{\boldsymbol{\xi}}\left(\tilde{\boldsymbol{z}}\right),\boldsymbol{c}'\right)+d\left(\boldsymbol{z},\tilde{\boldsymbol{z}}\right),
\end{aligned}
\tag{4}
$$

where $\beta\in(0,1]$ is the weight of the intervenability term. Note that for simplicity, we treat the probe's parameters $\boldsymbol{\xi}$ as fixed; however, since the outer optimisation problem is defined w.r.t. parameters $\boldsymbol{\phi}$, ideally, the probe would need to be optimised as the third, inner-most level of the problem. To avoid trilevel optimisation, we consider a special case of Equation 4 under $\beta=1$. For $\beta=1$, Equation 4 simplifies to $\min_{\boldsymbol{\psi},\boldsymbol{z}'}\mathbb{E}_{(\boldsymbol{x},\boldsymbol{c},y)\sim\mathcal{D},\,\boldsymbol{c}'\sim\pi}[\mathcal{L}^y(g_{\boldsymbol{\psi}}\left(\boldsymbol{z}'\right),y)]$, s.t. $\boldsymbol{z}'\in\arg\min_{\tilde{\boldsymbol{z}}}\lambda\mathcal{L}^{\boldsymbol{c}}\left(q_{\boldsymbol{\xi}}\left(\tilde{\boldsymbol{z}}\right),\boldsymbol{c}'\right)+d\left(\boldsymbol{z},\tilde{\boldsymbol{z}}\right)$. Thus, the parameters of $h_{\boldsymbol{\phi}}$ do not need to be optimised, and, hence, the probing function can be left fixed, as activations $\boldsymbol{z}$ are not affected by the fine-tuning. We consider this case to (i) computationally simplify the problem and (ii) keep the network's representations unchanged after fine-tuning for purposes of transfer learning for other downstream tasks.

## 4 EXPERIMENTAL SETUP

**Datasets** We evaluate the proposed methods on synthetic and real-world classification benchmarks summarised in Table C.1 (Appendix C). All datasets were divided according to the 60%-20%-20% train-validation-test split. Fine-tuning has been performed on the validation data, and evaluation on the test set. Further relevant details can be found in Appendix C.

For controlled experiments, we have adapted the nonlinear **synthetic** tabular dataset introduced by Marcinkevičs et al. (2023). Similar to Shin et al. (2023), we consider several data-generating mechanisms shown in Figure C.1, Appendix C.1. We refer to these three scenarios as *bottleneck*, *confounder*, and *incomplete*. The first scenario directly matches the inference graph of the vanilla CBM. The *confounder* is a setting wherein $\boldsymbol{c}$ and $\boldsymbol{x}$ are generated by an unobserved confounder and $y$ is generated by $\boldsymbol{c}$. Lastly, *incomplete* is a scenario with incomplete concepts, where $\boldsymbol{c}$ does not fully explain the association between $\boldsymbol{x}$ and $y$. Here, unexplained variance is modelled via the residual connection $\boldsymbol{x}\rightarrow y$. Another benchmark we consider is the **Animals with Attributes 2 (AwA2)** natural image dataset (Lampert et al., 2009; Xian et al., 2019). It includes animal images accompanied by 85 binary attributes and species labels. To further corroborate our findings, we perform experiments on the Caltech-UCSD Birds-200-2011 (**CUB**) dataset (Wah et al., 2011) (Appendix C.3), adapted for the CBM setting as described by Koh et al. (2020). We report these results in Appendix E.3.

Finally, to explore a practical setting, we utilise our techniques in chest radiograph classification. Namely, we test them on publicly available **CheXpert** (Irvin et al., 2019) and **MIMIC-CXR** (Johnson et al., 2019) datasets from the Stanford Hospital and Beth Israel Deaconess Medical Center, Boston, MA. Both datasets feature 14 binary attributes extracted from radiologist reports by automated labelling. In our analysis, the *Finding/No Finding* attribute is the target variable, and the

remaining labels are the concepts, similar to Chauhan et al. (2023). For simplicity, we retain a single X-ray per patient, excluding data with uncertain labels. Further details are in Appendix C.4.

**Baselines & Methods**  Below, we briefly outline the neural network models and fine-tuning techniques compared. All methods were implemented using PyTorch (v 1.12.1) (Paszke et al., 2019). Appendix D provides additional details. The code is available in an anonymised repository at https://anonymous.4open.science/r/intervenable-models-85E6/.

Firstly, we train a standard neural network (**BLACK BOX**) without concept knowledge, *i.e.* on the dataset of tuples $\{(\boldsymbol{x}_i, y_i)\}_i$. We utilise our technique for intervening post hoc by training a probe to predict concepts and editing the network's activations (Equation 1, Section 3.1). As an interpretable baseline, we consider the vanilla concept bottleneck model (**CBM**) by Koh et al. (2020). Across all experiments, we restrict ourselves to the joint bottleneck version, which minimises the weighted sum of the target and concept prediction losses: $\min_{\boldsymbol{\phi},\boldsymbol{\psi}} \mathbb{E}_{(\boldsymbol{x},\boldsymbol{c},y)\sim\mathcal{D}} [\mathcal{L}^y (f_{\boldsymbol{\theta}} (\boldsymbol{x}), y) + \alpha \mathcal{L}^{\boldsymbol{c}} (h_{\boldsymbol{\phi}} (\boldsymbol{x}), \boldsymbol{c})]$, where $\alpha > 0$ is a hyperparameter controlling the tradeoff between the two loss terms. Finally, as the primary method of interest, we apply our fine-tuning for intervenability technique (**FINE-TUNED, I**; Equation 4, Section 3.3) on the annotated validation set $\{(\boldsymbol{x}_i, \boldsymbol{c}_i, y_i)\}_i$.

As a common-sense baseline, we fine-tune the black box by training a probe to predict the concepts from intermediate representations (**FINE-TUNED, MT**). This amounts to multitask (MT) learning with hard weight sharing (Ruder, 2017). Specifically, the model is fine-tuned by minimising the following MT loss: $\min_{\boldsymbol{\phi},\boldsymbol{\psi},\boldsymbol{\xi}} \mathbb{E}_{(\boldsymbol{x},\boldsymbol{c},y)\sim\mathcal{D}} [\mathcal{L}^y (f_{\boldsymbol{\theta}} (\boldsymbol{x}), y) + \alpha \mathcal{L}^{\boldsymbol{c}} (q_{\boldsymbol{\xi}} (h_{\boldsymbol{\phi}} (\boldsymbol{x})), \boldsymbol{c})]$. As another baseline, we fine-tune the black box by appending concepts to the network's activations (**FINE-TUNED, A**). At test time, unknown concept values are set to $0.5$. To prevent overfitting and handle concept missingness, randomly chosen concept variables are masked during training. The objective is given by $\min_{\tilde{\boldsymbol{\psi}}} \mathbb{E}_{(\boldsymbol{x},\boldsymbol{c},y)\sim\mathcal{D}}[\mathcal{L}^y(\tilde{g}_{\tilde{\boldsymbol{\psi}}} ([h_{\boldsymbol{\phi}} (\boldsymbol{x}), \boldsymbol{c}]), y)]$, where $\tilde{g}$ takes as input concatenated activation and concept vectors. Note that, for this baseline, the parameters $\boldsymbol{\phi}$ remain fixed during fine-tuning. Last but not least, as a strong baseline resembling the approaches by Yuksekgonul et al. (2023) and Oikarinen et al. (2023), we train a CBM post hoc (**POST HOC CBM**) by solving the following problem: $\min_{\boldsymbol{\xi},\boldsymbol{\psi}} \mathbb{E}_{(\boldsymbol{x},\boldsymbol{c},y)\sim\mathcal{D}} [\mathcal{L}^y (g_{\boldsymbol{\psi}} (q_{\boldsymbol{\xi}} (h_{\boldsymbol{\phi}} (\boldsymbol{x}))), y) + \alpha \mathcal{L}^{\boldsymbol{c}} (q_{\boldsymbol{\xi}} (h_{\boldsymbol{\phi}} (\boldsymbol{x})), \boldsymbol{c})]$. The architectures of individual modules were kept as similar as possible for a fair comparison across all techniques.

**Evaluation**  To compare the proposed methods, we conduct interventions and analyze their performance under varying concept subset sizes. We report the areas under the receiver operating characteristic (AUROC) and precision-recall curves (AUPR) (Davis & Goadrich, 2006) since these performance measures provide a well-rounded summary over varying cutoff points and it might be challenging to choose a single cutoff in high-stakes decision areas. We utilise the Brier score (Brier, 1950) to gauge the accuracy of probabilistic predictions and, in addition, evaluate calibration.

## 5  RESULTS

**Results on Synthetic Data**  Figure 3 shows intervention results obtained across ten independent simulations under three generative mechanisms (Figure C.1, Appendix C.1) on the synthetic tabular data. Results w.r.t. the AUPR are very similar and can be found in Figure E.1, Appendix E.1. Across all three scenarios, we observe that, in principle, the proposed intervention procedure can improve the predictive performance of a black-box neural network. However, as expected, interventions are considerably more effective in CBMs than in untuned black-box classifiers: the former exhibit a steeper increase in performance given more ground-truth concept values. Generally, models explicitly fine-tuned for intervenability (FINE-TUNED, I) significantly improve over the original classifier, achieving intervention curves comparable to those of the CBM for the *bottleneck* and *incomplete* settings. Importantly, under an incomplete concept set (Figure 3(c)), black-box classifiers are expectedly superior to the ante hoc CBM, and fine-tuning for intervenability improves intervention effectiveness while maintaining the performance gap. Other fine-tuning strategies (FINE-TUNED, MT and FINE-TUNED, A) are either less effective or harmful, leading to a lower increase in AUROC and AUPR than attained by the untuned black box. Lastly, CBMs trained post hoc perform well in the *bottleneck* and *confounder* scenarios, being slightly less intervenable than FINE-TUNED, I. However, for the *incomplete* setting, interventions hurt the performance of the post hoc CBM. This behaviour may be related to the leakage described by Havasi et al. (2022).

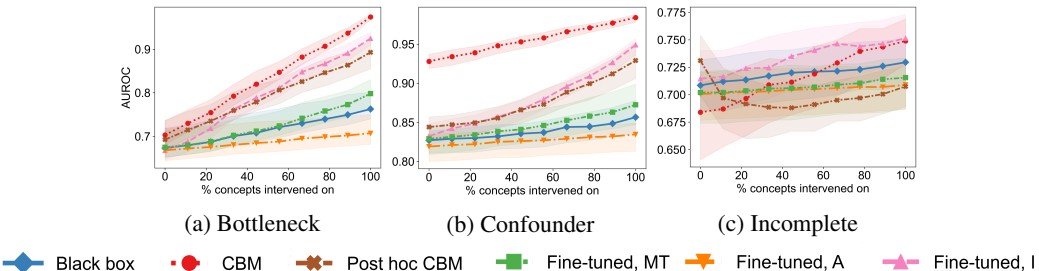


(a) Bottleneck      (b) Confounder      (c) Incomplete


Figure 3: Effectiveness of interventions w.r.t. target AUROC on the synthetic data under three generative mechanisms. Interventions were performed on the test set across ten independent simulations. Bold lines correspond to medians, and confidence bands are given by interquartile ranges.

In Table 1, we report the test-set performance of the models *without* interventions (under the *bottleneck* mechanism). For the concept prediction, expectedly, CBM outperforms black-box models, even after fine-tuning with the MT loss. However, without interventions, all models attain comparable AUROCs and AUPRs at the target prediction. Interestingly, fine-tuning for intervenability results in better-calibrated probabilistic predictions with lower Brier scores than those made by the original black box and after applying other fine-tuning strategies. As evidenced by Figure E.5(a) (Appendix E.4), fine-tuning has a regularising effect, reducing the false overconfidence observed in neural networks (Guo et al., 2017). Figure E.2 (Appendix E.1) contains further ablations for the intervention procedure on the influence of the hyperparameters, intervention strategies, and probe. In addition, Appendix E.2 explores the effect of interventions on the distribution of representations.

**Results on AwA2** Additionally, we explore the AwA2 dataset in Figure 4(a). This is a simple classification benchmark with class-wide concepts helpful for predicting the target. Hence, CBMs trained ante and post hoc are highly performant and intervenable. Nevertheless, untuned black-box models also benefit from concept-based interventions. In agreement with our findings on the synthetic dataset and in contrast to the other fine-tuning methods, ours enhances the performance of black-box models. Notably, black boxes fine-tuned for intervenability even surpass CBMs. Overall, the simplicity of this dataset leads to the generally high AUROCs and AUPRs across all methods.

To further investigate the impact of different hyperparameters on the interventions, we have performed ablation studies on untuned black-box models. These results are shown in Figures 4(b)–(d). Firstly, we vary the $\lambda$-parameter from Equation 3, which weighs the cross-entropy term, encouraging representation consistency with the given concept values. The results in Figure 4(b) suggest that interventions are effective across all $\lambda$s. Expectedly, higher hyperparameter values yield more effective interventions, *i.e.* a steeper increase in AUROC and AUPR. Figure 4(c) compares two intervention strategies: randomly selecting a concept subset (random) and prioritising the most uncertain concepts (uncertainty) (Shin et al., 2023) to intervene on (Algorithms D.1 and D.2, Appendix D). The intervention strategy has a clear impact on the performance increase, with the uncertainty-based approach yielding a steeper improvement. Finally, Figure 4(d) compares linear and nonlinear probes. Here, intervening via a nonlinear function leads to a significantly higher performance increase. Last but not least, Table 1 contains evaluation metrics at test time without interventions for target and concept prediction. We observe comparable performance across the methods, which are all successful due to the large dataset size and the relative simplicity of the classification task.

**Application to Chest X-ray Classification** To showcase the practicality of our approach, we present empirical findings on two chest X-ray datasets, MIMIC-CXR and CheXpert. Figure 5 shows intervention curves across ten independent initialisations. Interestingly, in both datasets, untuned black-box neural networks are not intervenable. By contrast, after fine-tuning for intervenability, the model's predictive performance and effectiveness of interventions improve visibly and even surpass those of the CBM. Given the challenging nature of these datasets, featuring instance-level concept labels, the final predictions by black boxes may not be as strongly reliant on the attributes, and CBMs do not necessarily outperform black-box networks, unlike in simpler benchmarking datasets. Finally, post hoc CBMs exhibit a behaviour similar to the synthetic dataset with incomplete concepts: interventions have no or even an adverse effect on performance.

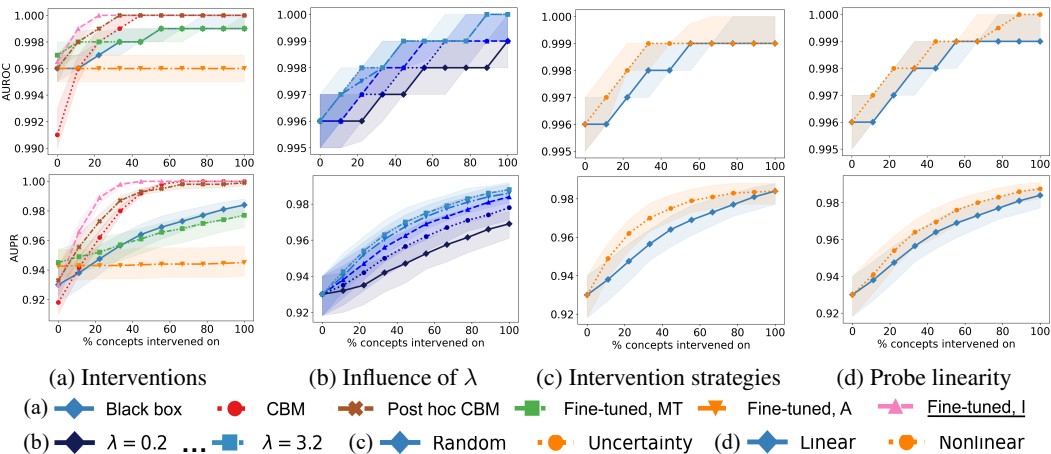

Figure 4: Intervention results on the AwA2 dataset w.r.t. target AUROC (*top*) and AUPR (*bottom*) across ten independent train-validation-test splits. (a) Comparison among considered methods. (b) Intervention results for the untuned black-box model under varying values of $\lambda \in \{0.2, 0.4, 0.8, 1.6, 3.2\}$ (Equation 3). **Darker** colours correspond to lower values. (c) Comparison between **random-subset** and **uncertainty-based** intervention strategies. (d) Comparison between **linear** and **nonlinear** probing functions.

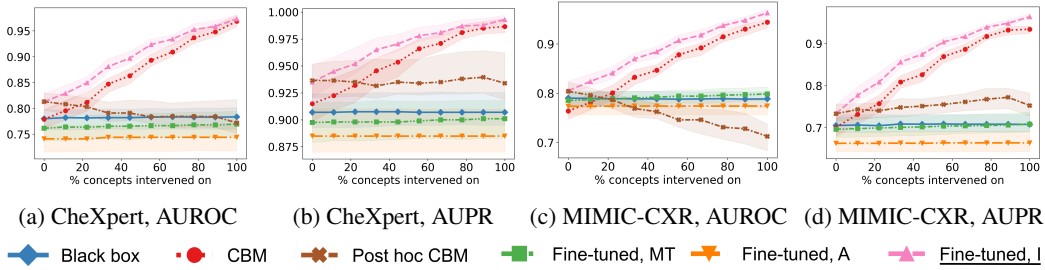

Figure 5: Intervention results w.r.t. target AUROC (a, c) and AUPR (b, d) across ten initialisations on the CheXpert (a, b) and MIMIC-CXR (c, d) datasets.

Furthermore, Table 1 shows the target and concept prediction performance without interventions. CBMs exhibit better performance for concept prediction, while fine-tuned models and post hoc CBMs outperform them at the target classification. Similar to the synthetic dataset, fine-tuning for intervenability enhances calibration, as evidenced Figure E.5 (Appendix E.4) and Brier scores.

## 6 DISCUSSION & CONCLUSION

This work has formalised intervenability as a measure of the effectiveness of concept-based interventions. It has also introduced techniques for performing instance-specific concept-based interventions on neural networks post hoc and fine-tuning the black-box models to improve their intervenability.

In contrast to interpretable models such as CBMs (Koh et al., 2020), our method circumvents the need for concept labels during training, which can be a substantial challenge in practical applications. Unlike recent works on converting black boxes into CBMs post hoc (Yuksekgonul et al., 2023; Oikarinen et al., 2023), we propose an effective intervention method that is faithful to the original architecture and representations. Our approach does not impose restrictions on the size of the bottleneck layer since it uses a probing function defined on the concept set and any chosen layer. Another line of work on CCEs (Abid et al., 2022) tackles a different problem—generating counterfactual explanations for the given prediction—using somewhat similar techniques. We have provided a detailed discussion of salient differences in Appendix B.

Table 1: Test-set concept and target prediction performance **without interventions** on the synthetic, AwA2, and chest X-ray datasets. For black-box models, concepts were predicted via a linear probe. Results are reported as averages and standard deviations across ten seeds. For concepts, performance metrics were averaged. Best results are reported in **bold**, second best are in *italics*.

| Dataset | Model | Concepts | | | Target | | |
|---|---|---|---|---|---|---|---|
| | | AUROC | AUPR | Brier | AUROC | AUPR | Brier |
| Synthetic | BLACK BOX | 0.716±0.018 | 0.710±0.017 | 0.208±0.006 | 0.686±0.043 | 0.675±0.046 | 0.460±0.003 |
| | CBM | **0.837±0.008** | **0.835±0.008** | *0.196±0.006* | **0.713±0.040** | **0.700±0.038** | 0.410±0.012 |
| | POST HOC CBM | 0.725±0.040 | 0.720±0.040 | 0.208±0.007 | *0.707±0.049* | *0.698±0.048* | *0.299±0.041* |
| | FINE-TUNED, A | — | — | — | 0.682±0.047 | 0.668±0.046 | 0.470±0.004 |
| | FINE-TUNED, MT | *0.784±0.013* | *0.780±0.014* | **0.186±0.006** | 0.687±0.046 | 0.668±0.043 | 0.471±0.003 |
| | FINE-TUNED, I | 0.716±0.018 | 0.710±0.017 | 0.208±0.006 | 0.695±0.051 | 0.685±0.051 | **0.285±0.014** |
| AwA2 | BLACK BOX | 0.991±0.002 | *0.979±0.006* | 0.027±0.006 | *0.996±0.001* | 0.926±0.020 | 0.199±0.038 |
| | CBM | *0.993±0.001* | 0.979±0.002 | *0.025±0.001* | 0.988±0.001 | 0.892±0.005 | 0.234±0.009 |
| | POST HOC CBM | 0.990±0.002 | 0.976±0.006 | 0.027±0.005 | *0.996±0.001* | *0.927±0.019* | *0.176±0.035* |
| | FINE-TUNED, A | — | — | — | *0.996±0.001* | **0.938±0.016** | **0.170±0.036** |
| | FINE-TUNED, MT | **0.994±0.002** | **0.985±0.004** | **0.022±0.005** | **0.997±0.001** | **0.938±0.017** | 0.178±0.038 |
| | FINE-TUNED, I | 0.991±0.002 | *0.979±0.005* | 0.027±0.006 | *0.996±0.001* | 0.925±0.020 | 0.195±0.040 |
| CheXpert | BLACK BOX | 0.665±0.003 | 0.257±0.003 | *0.097±0.001* | 0.785±0.011 | 0.911±0.006 | 0.305±0.009 |
| | CBM | **0.723±0.005** | **0.322±0.003** | 0.116±0.001 | 0.786±0.009 | 0.919±0.006 | 0.375±0.013 |
| | POST HOC CBM | 0.597±0.007 | 0.221±0.03 | 0.103±0.001 | **0.820±0.080** | **0.939±0.004** | *0.206±0.005* |
| | FINE-TUNED, A | — | — | — | 0.749±0.008 | 0.891±0.005 | 0.329±0.013 |
| | FINE-TUNED, MT | *0.684±0.003* | *0.275±0.003* | **0.094±0.001** | 0.768±0.019 | 0.901±0.012 | 0.297±0.012 |
| | FINE-TUNED, I | 0.668±0.004 | 0.257±0.003 | *0.097±0.001* | *0.819±0.009* | *0.938±0.004* | **0.201±0.007** |
| MIMIC-CXR | BLACK BOX | 0.743±0.006 | 0.170±0.004 | *0.046±0.001* | *0.789±0.006* | *0.706±0.009* | 0.444±0.003 |
| | CBM | *0.744±0.006* | **0.224±0.003** | 0.053±0.001 | 0.765±0.007 | 0.699±0.006 | 0.427±0.003 |
| | POST HOC CBM | 0.713±0.007 | 0.157±0.008 | *0.046±0.001* | **0.808±0.006** | **0.733±0.009** | **0.306±0.006** |
| | FINE-TUNED, A | — | — | — | 0.773±0.009 | 0.665±0.013 | 0.459±0.004 |
| | FINE-TUNED, MT | **0.748±0.008** | *0.187±0.003* | **0.045±0.001** | 0.785±0.006 | 0.696±0.009 | 0.450±0.008 |
| | FINE-TUNED, I | *0.744±0.005* | 0.172±0.005 | *0.046±0.001* | **0.808±0.007** | **0.733±0.009** | *0.314±0.015* |

Empirically, we demonstrated that black-box models trained without explicit concept knowledge are intervenable on synthetic tabular and natural image data, given an annotated validation set. We also showed that fine-tuning for intervenability improved the effectiveness of the interventions, bringing black boxes on par with CBMs, and led to better-calibrated predictions. Additionally, we explored the practical applicability of our techniques in chest X-ray classification. In this more realistic setting, black-box classifiers were not directly intervenable. However, the proposed fine-tuning procedure alleviated this limitation. On simpler benchmarks, where concepts are directly embedded into the data-generating mechanism and are sufficient (Yeh et al., 2020), CBMs slightly outperformed all variants of black-box models and post hoc CBMs. By contrast, in a more realistic setting, where concepts differ at the instance level or are incomplete, black-box models fine-tuned for intervenability surpass CBMs. Moreover, interventions on a CBM trained post hoc in that setting are not effective or are even harmful. Lastly, in addition to the findings above, we introduced and studied common-sense fine-tuning baselines that performed worse than the proposed method, highlighting the need for explicitly including the intervenability in the loss function.

To summarise, this work has explored the tradeoffs between interpretability, intervenability, and performance in black-box predictive models. In particular, we have proposed procedures allowing for effective concept-based interventions without a need to train CBMs ante hoc or alter the model's architecture or representations while only requiring concept labels for probing and fine-tuning.

**Limitations & Future Work** The current work opens many avenues for future research and improvements. Firstly, the variant of the fine-tuning procedure considered in this paper does not affect the neural network's representations. However, it would be interesting to investigate a more general formulation wherein all model and probe parameters are fine-tuned end-to-end. According to our empirical findings, the choice of intervention strategy, hyperparameters, and probing function can influence the effectiveness of interventions. A more in-depth experimental investigation of these aspects is warranted. Furthermore, we only considered having a single fixed intervention strategy throughout fine-tuning, whereas further improvement could come from learning an optimal strategy alongside fine-tuned weights. Lastly, the proposed techniques rely on the annotated validation data to fit a probe and could benefit from (semi-)automated concept discovery, *e.g.* using multimodal models.

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

APPENDIX

## A  FINE-TUNING FOR INTERVENABILITY

Algorithm A.1 contains the detailed pseudocode for fine-tuning for intervenability described in Section 3.3. Recall that the black-box model $f_{\boldsymbol{\theta}}$ is fine-tuned using a combination of the target prediction loss and intervenability defined in Equation 3. The implementation below applies to the special case of $\beta = 1$, which leads to the simplified loss. Importantly, in this case, the parameters $\phi$ are treated as fixed, and the probing function $q_{\boldsymbol{\xi}}$ does not need to be fine-tuned alongside the model. Lastly, note that, in Algorithm A.1, interventions are performed for whole batches of data points $\boldsymbol{x}_b$ using the procedure described in Section 3.1.

---

**Algorithm A.1:** Fine-tuning for Intervenability

**Input:** Trained black-box model $f_{\boldsymbol{\theta}} = \langle g_{\boldsymbol{\psi}}, h_{\boldsymbol{\phi}} \rangle$, probing function $q_{\boldsymbol{\xi}}$, concept prediction loss function $\mathcal{L}^c$, target prediction loss function $\mathcal{L}^y$, validation set $\{(\boldsymbol{x}_i, \boldsymbol{c}_i, y_i)\}_{i=1}^{N}$, intervention strategy $\pi$, distance function $d$, hyperparameter value $\lambda > 0$, maximum number of steps $E_I$ for the intervention procedure, parameter for the convergence criterion $\varepsilon_I > 0$ for the intervention procedure, learning rate $\eta_I > 0$ for the intervention procedure, number of fine-tuning epochs $E$, mini-batch size $M$, learning rate $\eta > 0$

**Output:** Fine-tuned model

1   Train the probing function $q_{\boldsymbol{\xi}}$ on the validation set,

    *i.e.* $\boldsymbol{\xi} \leftarrow \arg\min_{\boldsymbol{\xi}'} \sum_{i=1}^{N} \mathcal{L}^c\left(q_{\boldsymbol{\xi}'}\left(h_{\boldsymbol{\phi}}\left(\boldsymbol{x}_i\right)\right), \boldsymbol{c}_i\right)$          ▷ Step 1: Probing

2   **for** $e = 0$ *to* $E - 1$ **do**

3      Randomly split $\{1, ..., N\}$ into mini-batches of size $M$ given by $\mathcal{B}$

4      **for** $b \in \mathcal{B}$ **do**

5         $\boldsymbol{z}_b \leftarrow h_{\boldsymbol{\phi}}\left(\boldsymbol{x}_b\right)$

6         $\hat{y}_b \leftarrow g_{\boldsymbol{\psi}}\left(\boldsymbol{z}_b\right)$

7         $\hat{\boldsymbol{c}}_b \leftarrow q_{\boldsymbol{\xi}}\left(\boldsymbol{z}_b\right)$

8         Sample $\boldsymbol{c}_b' \sim \pi$

9         Initialise $\boldsymbol{z}_b' = \boldsymbol{z}_b$, $\boldsymbol{z}_{b,\text{old}}' = \boldsymbol{z}_b + \varepsilon_I \boldsymbol{e}$, and $e_I = 0$     ▷ Step 2: Editing Representations

10         **while** $\left\| \boldsymbol{z}_b' - \boldsymbol{z}_{b,\text{old}}' \right\|_1 \geq \varepsilon_I$ ***and*** $e_I < E_I$ **do**

11            $\boldsymbol{z}_{b,\text{old}}' \leftarrow \boldsymbol{z}_b'$

12            $\boldsymbol{z}_b' \leftarrow \boldsymbol{z}_b' - \eta_I \nabla_{\boldsymbol{z}_b'} \left[ d(\boldsymbol{z}_b, \boldsymbol{z}_b') + \lambda \mathcal{L}^c\left(q_{\boldsymbol{\xi}}\left(\boldsymbol{z}_b'\right), \boldsymbol{c}_b'\right) \right]$       ▷ Equation 1

13            $e_I \leftarrow e_I + 1$

14         **end**

15         $\hat{y}_b' \leftarrow g_{\boldsymbol{\psi}}\left(\boldsymbol{z}_b'\right)$            ▷ Step 3: Updating Output

16         $\boldsymbol{\psi} \leftarrow \boldsymbol{\psi} - \eta \nabla_{\boldsymbol{\psi}} \mathcal{L}^y\left(\hat{y}_b', y_b\right)$            ▷ Equation 4

17      **end**

18   **end**

19   **return** $f_{\boldsymbol{\theta}}$

---

## B    FURTHER REMARKS & DISCUSSION

This appendix contains extended remarks and discussion beyond the scope of the main text.

**Design Choices for the Intervention Procedure**    The intervention procedure entails a few design choices, including the (non)linearity of the probing function, the distance function in the objective from Equation 1, and the tradeoff between consistency and proximity determined by $\lambda$ from Equation 1. We have explored some of these choices empirically in our ablation experiments (see Figure 4 and Appendix E). Naturally, interventions performed on black-box models using our method are meaningful in so far as the activations of the neural network are correlated with the given high-level attributes and the probing function $q_{\boldsymbol{\xi}}$ can be trained to predict these attribute values accurately. Otherwise, edited representations and updated predictions are likely to be spurious and may harm the model's performance.

**Should All Models Be Intervenable?**    Intervenability (Equation 3), in combination with the probing function, can be used to evaluate the interpretability of a black-box predictive model and help understand whether (i) learnt representations capture information about given human-understandable attributes and whether (ii) the network utilises these attributes and can be interacted with. However, a black-box model does not always need to be intervenable. For instance, when the given concept set is not predictive of the target variable, the black box trained using supervised learning should not and probably would not rely on the concepts. On the other hand, if the model's representations are nearly perfectly correlated with the attributes, providing the ground truth should not significantly impact the target prediction loss. Lastly, the model's intervenability may depend on the chosen intervention strategy, which may not always lead to the expected decrease in the loss.

**Intervenability & Variable Importance**    As mentioned in Section 3.2, intervenability (Equation 2) measures the effectiveness of interventions performed on a model by quantifying a gap between the expected target prediction loss with and without performing concept-based interventions. Equation 2 is reminiscent of the model reliance (MR) (Fisher et al., 2019) used for quantifying variable importance.

Informally, for a predictive model $f$, MR measures the importance of some feature of interest and is defined as

$$MR(f) := \frac{\text{expected loss of } f \text{ under noise}}{\text{expected loss of } f \text{ without noise}}. \tag{B.1}$$

Above, the noise augments the inputs of $f$ and must render the feature of interest uninformative of the target variable. One practical instance of the model reliance is permutation-based variable importance (Breiman, 2001; Molnar, 2022).

The intervenability measure in Equation 2 can be summarised informally as the *difference* between the expected loss of $g_{\psi}$ without interventions and the loss under interventions. Suppose intervention strategy $\pi$ is specified so that it augments a single concept in $\hat{\boldsymbol{c}}$ with noise (Equation B.1). In that case, intervenability can be used to quantify the reliance of $g_{\psi}$ on the concept variable of interest in $\hat{\boldsymbol{c}}$. The main difference is that Equation B.1 is given by the ratio of the expected losses, whereas intervenability looks at the difference of expectations.

**Comparison with Conceptual Counterfactual Explanations**    We can draw a relationship between the concept-based interventions (Equation 3) and conceptual counterfactual explanations (CCE) studied by Abid et al. (2022) and Kim et al. (2023b). In brief, interventions aim to "inject" concepts $\boldsymbol{c}'$ provided by the user into the network's representation to affect and improve the downstream prediction. By contrast, CCEs seek to identify a sparse set of concept variables that could be leveraged to flip the label predicted by the classifier $f_{\boldsymbol{\theta}}$. Thus, the problem tackled in the current work is different from and complementary to CCE.

More formally, following the notation from Section 1, a conceptual counterfactual explanation (Abid et al., 2022) is given by

$$\arg\min_{\boldsymbol{w}} \mathcal{L}^y \left( g_{\psi} \left( h_{\boldsymbol{\phi}}\left(\boldsymbol{x}\right) + \boldsymbol{w}\tilde{\boldsymbol{C}} \right), y' \right) + \alpha \left\| \boldsymbol{w} \right\|_1 + \beta \left\| \boldsymbol{w} \right\|_2,$$
$$\text{s.t. } \boldsymbol{w}^{\min} \leq \boldsymbol{w} \leq \boldsymbol{w}^{\max}, \tag{B.2}$$

where $\tilde{C}$ is the concept bank, $y'$ is the given target value (in classification, the opposite to the predicted $\hat{y}$), $\alpha, \beta > 0$ are penalty weights, and $\left[\boldsymbol{w}^{\text{min}}, \boldsymbol{w}^{\text{max}}\right]$ defines the desired range for weights $\boldsymbol{w}$. Note that further detailed constraints are imposed via the definition of $\left[\boldsymbol{w}^{\text{min}}, \boldsymbol{w}^{\text{max}}\right]$ in the original work by Abid et al. (2022).

Observe that the optimisation problem in Equation B.2 is defined w.r.t. the flipped label $y'$ and does not incorporate user-specified concepts $c'$ as opposed to interventions in Equation 1. Thus, CCEs aim to identify the concept variables that need to be "added" to flip the label output by the classifier. In contrast, interventions seek to perturb representations consistently with the *given* concept values.

## C  DATASETS

Below, we present further details about the datasets and preprocessing involved in the experiments (Section 4). The synthetic data can be generated using our code;[1] AwA2, CUB, CheXpert, and MIMIC-CXR datasets are publicly available. Table C.1 provides a brief summary of the datasets.

Table C.1: Dataset summary. After any filtering or preprocessing, $N$ is the total number of data points; $p$ is the input dimensionality; and $K$ is the number of concept variables.

| Dataset | Data type | $N$ | $p$ | $K$ |
|---|---|---|---|---|
| Synthetic | Tabular | 50,000 | 1,500 | 30 |
| AwA2 | Image | 37,322 | 224×224 | 85 |
| CUB | Image | 11,788 | 224×224 | 112 |
| CheXpert | Image | 49,408 | 224×224 | 13 |
| MIMIC-CXR | Image | 54,276 | 224×224 | 13 |

### C.1  SYNTHETIC TABULAR DATA

As mentioned in Section 4, to perform experiments in a controlled manner, we generate synthetic nonlinear tabular data using the procedure adapted from Marcinkevičs et al. (2023). We explore three settings corresponding to different data-generating mechanisms (Figure C.1): (a) *bottleneck*, (b) *confounder*, and (c) *incomplete*. The first scenario directly matches the inference graph of the vanilla CBM (Koh et al., 2020). The *confounder* is a setting wherein $c$ and $x$ are generated by an unobserved confounder $z$ and $y$ is generated by $c$. Lastly, *incomplete* is a scenario with incomplete concepts, where $c$ does not fully explain the variance in $y$. Here, unexplained variance is modelled as a latent variable $r$ via the path $x \rightarrow r \rightarrow y$. Unless mentioned otherwise, we mainly focus on the simplest scenario shown in Figure C.1(a). Below, we outline each generative process in detail. Throughout this appendix, let $N$, $p$, and $K$ denote the number of independent data points $\{(\boldsymbol{x}_i, \boldsymbol{c}_i, y_i)\}_{i=1}^{N}$, covariates, and concepts, respectively. Across all experiments, we set $N = 50{,}000$, $p = 1{,}500$, and $K = 30$.

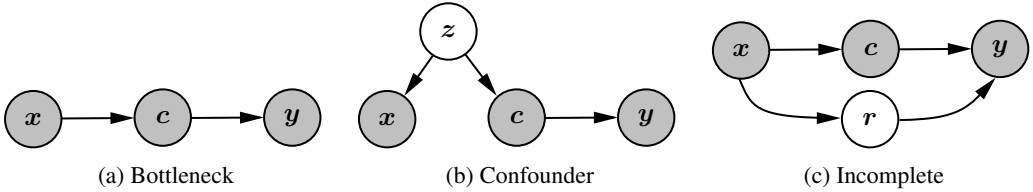

| (a) Bottleneck | (b) Confounder | (c) Incomplete |

Figure C.1: Data-generating mechanisms for the synthetic dataset summarised as graphical models. Each node corresponds to a random variable. Observed variables are shown in grey.

**Bottleneck**   In this setting, the covariates $\boldsymbol{x}_i$ generate binary-valued concepts $\boldsymbol{c}_i \in \{0, 1\}^{K}$, and the binary-valued target $y_i$ depends on the covariates exclusively via the concepts. The generative process is as follows:

1. Randomly sample $\boldsymbol{\mu} \in \mathbb{R}^p$ s.t. $\mu_j \sim \text{Uniform}\,(-5,\,5)$ for $1 \leq j \leq p$.

2. Generate a random symmetric, positive-definite matrix $\boldsymbol{\Sigma} \in \mathbb{R}^{p \times p}$.

3. Randomly sample a design matrix $\boldsymbol{X} \in \mathbb{R}^{N \times p}$ s.t. $\boldsymbol{X}_{i,:} \sim \mathcal{N}_p\,(\boldsymbol{\mu},\,\boldsymbol{\Sigma})$.[2]

4. Let $h: \mathbb{R}^p \rightarrow \mathbb{R}^K$ and $g: \mathbb{R}^K \rightarrow \mathbb{R}$ be randomly initialised multilayer perceptrons with ReLU nonlinearities.

5. Let $c_{i,k} = \mathbf{1}_{\left\{[h(\boldsymbol{X}_{i,:})]_k \geq m_k\right\}}$, where $m_k = \text{median}\left(\left\{[h\,(\boldsymbol{X}_{l,:})]_k\right\}_{l=1}^{N}\right)$, for $1 \leq i \leq N$ and $1 \leq k \leq K$.

6. Let $y_i = \mathbf{1}_{\{g(\boldsymbol{c}_i) \geq m_y\}}$, where $m_y = \text{median}\left(\{g\,(\boldsymbol{c}_i)\}_{l=1}^{N}\right)$, for $1 \leq i \leq N$.

---

[1] https://anonymous.4open.science/r/intervenable-models-85E6/

[2] $\boldsymbol{X}_{i,:}$ refers to the $i$-th row of the design matrix, *i.e.* the covariate vector $\boldsymbol{x}_i$

**Confounder**   Another scenario we consider is where $x$ and $c$ are generated by an unobserved confounder:

1. Randomly sample $Z \in \mathbb{R}^{N \times K}$ s.t. $z_{i,k} \sim \mathcal{N}(0,1)$ for $1 \leq i \leq N$ and $1 \leq k \leq K$.

2. Let $c_{i,k} = \mathbf{1}_{\{z_{i,k} \geq 0\}}$ for $1 \leq i \leq N$ and $1 \leq k \leq K$.

3. Let $h : \mathbb{R}^K \to \mathbb{R}^p$ and $g : \mathbb{R}^K \to \mathbb{R}$ be randomly initialised multilayer perceptrons with ReLU nonlinearities.

4. Let $x_i = h(Z_{i,:})$ for $1 \leq i \leq N$.

5. Let $y_i = \mathbf{1}_{\{\sigma(g(c_i)) \geq 1/2)\}}$ for $1 \leq i \leq N$, where $\sigma$ denotes the sigmoid function.

**Incomplete**   Last but not least, to simulate the incomplete concept set scenario, where a part of concepts are latent, we slightly adjust the procedure from the *bottleneck* setting above:

1. Follow steps 1–3 from the *bottleneck* procedure.

2. Let $h : \mathbb{R}^p \to \mathbb{R}^{K+J}$ and $g : \mathbb{R}^{K+J} \to \mathbb{R}$ be randomly initialised multilayer perceptrons with ReLU nonlinearities, where $J$ is the number of unobserved concept variables.

3. Let $u_{i,k} = \mathbf{1}_{\left\{[h(X_{i,:})]_k \geq m_k\right\}}$, where $m_k = \mathrm{median}\left(\left\{[h(X_{l,:})]_k\right\}_{l=1}^N\right)$, for $1 \leq i \leq N$ and $1 \leq k \leq K+J$.

4. Let $c_i = u_{i,1:K}$ and $r_i = u_{i,(K+1):(K+J)}$ for $1 \leq i \leq N$.

5. Let $y_i = \mathbf{1}_{\{g(u_i) \geq m_y\}}$, where $m_y = \mathrm{median}\left(\{g(u_i)\}_{l=1}^N\right)$, for $1 \leq i \leq N$.

Note that, in steps 3–5 above, $u_i$ corresponds to the concatenation of $c_i$ and $r_i$. Across all experiments, we set $J = 90$.

## C.2   ANIMALS WITH ATTRIBUTES 2

Animals with Attributes 2[3] dataset (Lampert et al., 2009; Xian et al., 2019) serves as a natural image benchmark in our experiments. It comprises 37,322 images of 50 animal classes (species), each associated with 85 binary attributes utilised as concepts. An apparent limitation of this dataset is that the concept labels are shared across whole classes, similar to the Caltech-UCSD Birds experiment from the original work by Koh et al. (2020). Thus, AwA2 offers a simplified setting for transfer learning across different classes and is designed to address attribute-based classification and zero-shot learning challenges. In our evaluation, we used all the images in the dataset without any specialised preprocessing or preselection. All images were rescaled to $224 \times 224$ pixels.

## C.3   CALTECH-UCSD BIRDS

Caltech-UCSD Birds-200-2011[4] dataset (Wah et al., 2011) is another natural image benchmark explored in the original work on CBMs by Koh et al. (2020). It consists of 11,788 bird photographs from 200 species (classes) and originally includes 312 concepts, such as wing colour, beak shape, etc. We have followed the preprocessing routine proposed by Koh et al. (2020). Particularly, the final dataset includes only the 112 most prevalent binary attributes. We have included image augmentations during training, such as random horizontal flips, adjustments of the brightness and saturation, and normalisation. Similar to AwA2, CUB concepts are shared across all instances of individual classes. No additional specialised preprocessing was performed on the images, which were rescaled to a resolution of $224 \times 224$ pixels.

## C.4   CHEST X-RAY DATASETS

As mentioned, we conducted an empirical evaluation on two real-world chest X-ray datasets: CheXpert (Irvin et al., 2019) and MIMIC-CXR (Johnson et al., 2019). The former includes over 220,000

---

[3]https://cvml.ista.ac.at/AwA2/
[4]https://www.vision.caltech.edu/datasets/cub_200_2011/

chest radiographs from 65,240 patients at the Stanford Hospital.[5] These images are accompanied by 14 binary attributes extracted from radiologist reports using the CheXpert labeller (Irvin et al., 2019), a model trained to predict these attributes. MIMIC-CXR is another publicly available dataset containing chest radiographs in DICOM format, paired with free-text radiology reports.[6] It comprises more than 370,000 images associated with 227,835 radiographic studies conducted at the Beth Israel Deaconess Medical Center, Boston, MA, involving 65,379 patients. Similar to CheXpert, the same labeller was employed to extract the same set of 14 binary labels from the text reports. Notably, some concepts may be labelled as uncertain. Similar to Chauhan et al. (2023), we designate the *Finding/No Finding* attribute as the target variable for classification and utilise the remaining labels as concepts. In our implementation, we remove all the samples that contain uncertain labels and we discard multiple visits of the same patient, keeping only the last acquired recording per subject for both datasets. All images were cropped to a square aspect ratio and rescaled to $224 \times 224$ pixels. Additionally, augmentations were applied during training, namely, random affine transformations, including rotation up to 5 degrees, translation up to 5% of the image's width and height, and shearing with a maximum angle of 5 degrees. We also include a random horizontal flip augmentation to introduce variation in the orientation of recordings within the dataset.

---

[5] https://stanfordmlgroup.github.io/competitions/chexpert/
[6] https://physionet.org/content/mimic-cxr/2.0.0/

# D  IMPLEMENTATION DETAILS

This section provides implementation details, such as network architectures and intervention and fine-tuning procedure hyperparameter configurations. All models and procedures were implemented using PyTorch (v 1.12.1) (Paszke et al., 2019) and scikit-learn (v 1.0.2) (Pedregosa et al., 2011).

**Network & Probe Architectures**  For the synthetic tabular data, we utilise a fully connected neural network (FCNN) as the black-box model. Its architecture is summarised in Table D.1 in PyTorch-like pseudocode. For this classifier, probing functions are trained and interventions are performed on the activations of the third layer, *i.e.* the output after line **2** in Table D.1. For natural and medical image datasets, we use the ResNet-18 (He et al., 2016) with random initialisation followed by four fully connected layers and the sigmoid or softmax activation. Probing and interventions are performed on the activations of the second layer after the ResNet-18 backbone. For the CBMs, to facilitate fair comparison, we use the same architectures with the exception that the layers mentioned above were converted into bottlenecks with appropriate dimensionality and activation functions. Similar settings are used for post hoc CBMs with the addition of a linear layer mapping representations to the concepts.

For fine-tuning, we utilise a single fully connected layer with an appropriate activation function as a linear probe and a multilayer perceptron with a single hidden layer as a nonlinear function. For evaluation on the test set (Table 1), we fit a logistic regression classifier from scikit-learn as a linear probe. The logistic regression is only used for evaluation purposes and not interventions.

Table D.1: Fully connected neural network architecture used as a black-box classifier in the experiments on the synthetic tabular data. `nn` stands for `torch.nn`; F stands for `torch.nn.functional`; `input_dim` corresponds to the number of input features.

|   | **FCNN Classifier** |
|---|---|
| **1** | `nn.Linear(input_dim, 256)` |
|   | `F.relu()` |
|   | `nn.Dropout(0.05)` |
|   | `nn.BatchNorm1d(256)` |
| **2** | `for l in range(2):` |
|   | `    nn.Linear(256, 256)` |
|   | `    F.relu()` |
|   | `    nn.Dropout(0.05)` |
|   | `    nn.BatchNorm1d(256)` |
| **3** | `out = nn.Linear(256, 1)` |
| **4** | `torch.sigmoid()` |

**Interventions**  Unless mentioned otherwise, interventions on black-box models were performed using linear probes, the random-subset intervention strategy, and under $\lambda = 0.8$ (Equation 1). Recall that Figures 4 and E.2 provide ablation results on the influence of this hyperparameter. Despite some variability, the analysis shows that higher values of $\lambda$ expectedly lead to more effective interventions. The choice of $\lambda$ for our experiments was meant to represent the "average case", and no tuning was performed for this hyperparameter.

Similarly, we have mainly used a linear probing function and the simple random-subset intervention strategy to provide proof-of-concept results without extensive optimisation of the intervention strategy or the need for nonlinear probing. Thus, our primary focus was on demonstrating the intervenability of black-box models and showcasing the effectiveness of the fine-tuning method rather than an exhaustive hyperparameter search.

**Intervention Strategies**  In ablation studies, we compare two intervention strategies (Figure 4) inspired by Shin et al. (2023): (i) random-subset and (ii) uncertainty-based. Herein, we provide a more formal definition of these procedures described as pseudocode in Algorithms D.1–D.2. Recall that given a data point $(\boldsymbol{x}, \boldsymbol{c}, y)$ and predicted values $\hat{\boldsymbol{c}}$ and $\hat{y}$, an intervention strategy defines a distribution over intervened concept values $\boldsymbol{c}'$. Random-subset strategy (Algorithm D.1) replaces predicted values with the ground truth for several concept variables ($k$) chosen uniformly at random. By contrast, the uncertainty-based strategy (Algorithm D.2) samples concept variables to be replaced with the ground-truth values without replacement with initial probabilities proportional to

the concept prediction uncertainties, denoted by $\boldsymbol{\sigma}$. In our experiments, the components of $\hat{\boldsymbol{c}}$ are the outputs of the sigmoid function, and the uncertainties are computed as $\sigma_i = 1/\left(|\hat{c}_i - 0.5| + \varepsilon\right)$ (Shin et al., 2023) for $1 \leq i \leq K$, where $\varepsilon > 0$ is small.

---

**Algorithm D.1:** Random-subset Intervention Strategy

---

**Input:** A data point $(\boldsymbol{x}, \boldsymbol{c}, y)$, predicted concept values $\hat{\boldsymbol{c}}$, the number of concept variables to be intervened on $1 \leq k \leq K$
**Output:** Intervened concept values $\boldsymbol{c}'$

1 $\boldsymbol{c}' \leftarrow \hat{\boldsymbol{c}}$
2 Sample $\mathcal{I}$ uniformly at random from $\{\mathcal{S} \subseteq \{1, \ldots, K\} : |\mathcal{S}| = k\}$
3 $\boldsymbol{c}'_{\mathcal{I}} \leftarrow \boldsymbol{c}_{\mathcal{I}}$

4 **return** $\boldsymbol{c}'$

---

**Algorithm D.2:** Uncertainty-based Intervention Strategy

---

**Input:** A data point $(\boldsymbol{x}, \boldsymbol{c}, y)$, predicted concept values $\hat{\boldsymbol{c}}$, the number of concept variables to be intervened on $1 \leq k \leq K$
**Output:** Intervened concept values $\boldsymbol{c}'$

1 $\sigma_j \leftarrow 1/\left(|\hat{c}_j - 0.5| + \varepsilon\right)$ for $1 \leq j \leq K$, where $\varepsilon > 0$ is small
2 $\boldsymbol{\sigma} \leftarrow (\sigma_1 \quad \cdots \quad \sigma_K)$
3 $\boldsymbol{c}' \leftarrow \hat{\boldsymbol{c}}$
4 Sample $k$ indices $\mathcal{I} = \{i_j\}_{j=1}^{k}$ s.t. each $i_j$ is sampled without replacement from $\{1, \ldots, K\}$ with initial probabilities given by $(\boldsymbol{\sigma} + \varepsilon) / \left(K\varepsilon + \sum_{i=1}^{K} \sigma_i\right)$, where $\varepsilon > 0$ is small
5 $\boldsymbol{c}'_{\mathcal{I}} \leftarrow \boldsymbol{c}_{\mathcal{I}}$

6 **return** $\boldsymbol{c}'$

---

**Fine-tuning for Intervenability**  The fine-tuning procedure outlined in Section 3.3 and detailed in Algorithm A.1 necessitates intervening on the representations throughout the optimisation. During fine-tuning, we utilise the random-subset intervention strategy, *i.e.* interventions are performed on a subset of the concept variables by providing the ground-truth values. More concretely, interventions are performed on 50% of the concept variables chosen uniformly at random.

**Fine-tuning Baselines**  The baseline methods described in Section 4 incorporate concept information in distinct ways. On the one hand, the multitask learning approach, FINE-TUNED, MT, utilises the entire batch of concepts at each iteration during fine-tuning. For this procedure, we set $\alpha = 1.0$ (recall that $\alpha$ controls the tradeoff between the target and concept prediction loss terms). On the other hand, the FINE-TUNED, A approach, which appends the concepts to the network's activations, does not use the complete concept set for each batch. In particular, before appending, concept values are randomly masked and set to $0.5$ with a probability of $0.5$. This practical trick is reminiscent of the dropout (Srivastava et al., 2014) and is meant to help the model remain intervenable and handle missing concept values.

**Hyperparameters**  Below, we list key hyperparameter configurations; the remaining details are documented in our code. For the synthetic data, CBMs and black-box classifiers are trained for 150 and 100 epochs, respectively, with a learning rate of $10^{-4}$ and a batch size of 64. Across all other experiments, CBMs are trained for 350 epochs and black-box models for 300 epochs with a learning rate of $10^{-4}$ halved midway through training and a batch size of 64. CBMs are trained using the joint optimisation procedure (Koh et al., 2020) under $\alpha = 1.0$, where $\alpha$ controls the tradeoff between the concept and target prediction losses. All probes were trained on the validation data for 150 epochs with a learning rate of $10^{-2}$ using the stochastic gradient descent (SGD) optimiser. Finally, all fine-tuning procedures were run for 150 epochs with a learning rate of $10^{-4}$ and a batch size of 64 using the Adam optimiser. At test time, interventions were performed on batches of 512 data points.

# E  FURTHER RESULTS

This section contains supplementary results and ablation experiments not included in the main body of the text.

## E.1  FURTHER RESULTS ON SYNTHETIC DATA

Figure E.1 supplements the intervention experiment results in Figure 3, Section 5, showing intervention curves w.r.t. AUPR under the three generative mechanisms for the synthetic data. The overall patterns and conclusions are similar to those observed w.r.t. AUROC (Figure 3).

Figure E.2 provides ablation experiment results obtained on the synthetic tabular data under the *bottleneck* generative mechanism shown in Figure C.1(a), similar to the results reported for AwA2 in Figure 4, Section 5. In Figure E.2(a), we plot black-box intervention results across varying values of the hyperparameter $\lambda$ (Equation 1). As for AwA2, higher $\lambda$s result in more effective interventions: this finding is expected since $\lambda$ is the weight of the term penalising the inconsistency of the concept values predicted by the probe with the given values and, in the current implementation, interventions are performed using the ground truth. Interestingly, in Figure E.2(b), we observe no difference between the random subset and uncertainty-based intervention strategies. This could be explained by the fact that, in the synthetic dataset, all concepts by design are, on average, equally hard to predict and equally helpful in predicting the target variable (see Appendix C.1). Hence, the entropy-based uncertainty score should not be as informative in this dataset, and the order of intervention on the concepts should have little effect. Finally, similar to the main text, Figure E.2(c) suggests that a nonlinear probing function improves intervention effectiveness.

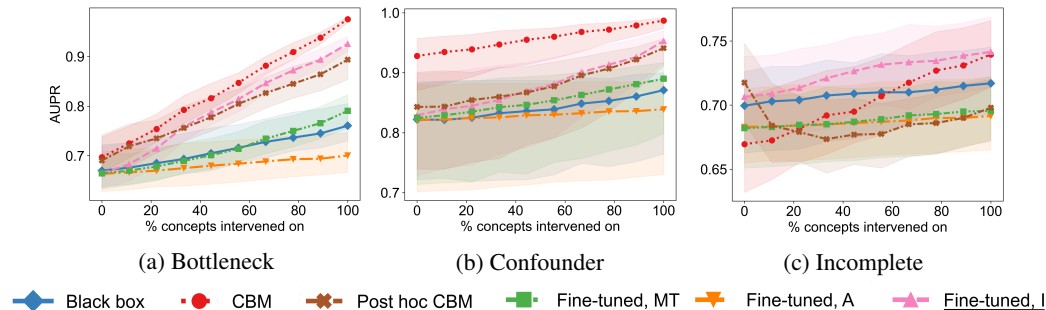

(a) Bottleneck     (b) Confounder     (c) Incomplete

Figure E.1: Effectiveness of interventions w.r.t. target AUPR on the synthetic tabular data under three different data-generating mechanisms (Figure C.1, Appendix C.1). Interventions were performed on the test set across ten independent simulations. Bold lines correspond to medians, and confidence bands are given by interquartile ranges.

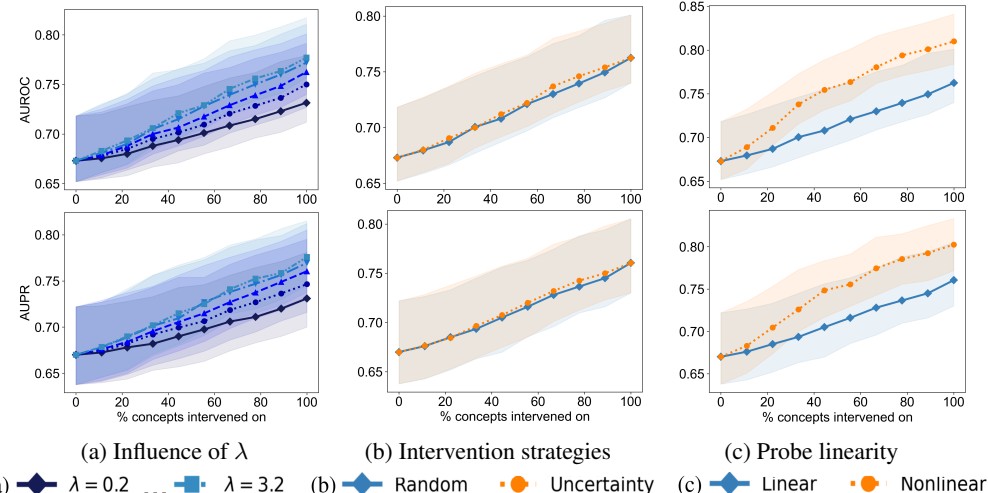

Figure E.2: Ablation study results w.r.t. target AUROC (*top*) and AUPR (*bottom*) on the synthetic dataset. Bold lines correspond to medians, and confidence bands are given by interquartile ranges across ten independent simulations. (a) Intervention results for the untuned black-box model under varying values of $\lambda \in \{0.2, 0.4, 0.8, 1.6, 3.2\}$ (Equation 3). **Darker** colours correspond to lower values. (b) Comparison between **random-subset** and **uncertainty-based** intervention strategies. (c) Comparison between **linear** and **nonlinear** probing functions.

## E.2   Effect of Interventions on Representations

In some cases (Abid et al., 2022), it may be deemed desirable that intervened representations $z'$ (Equation 1) remain plausible, i.e. their distribution should be close to that of the original representations $z$. Figure E.3 shows the first two principal components (PC) obtained from a batch of original and intervened representations from the synthetic dataset (under the *bottleneck* scenario) for two different values of the $\lambda$-parameter. We observe that, under $\lambda = 0.2$ (Figure E.3(a)), interventions affect representations, but $z'$ mainly stay close to $z$ w.r.t. the two PCs. By contrast, under $\lambda = 0.4$, interventions lead to a visible distribution shift, with many vectors $z'$ lying outside of the mass of $z$. This behaviour is expected since $\lambda$ controls the tradeoff between the consistency with the given concepts $c'$ and proximity. Thus, if the plausibility of intervened representations is desirable, the parameter $\lambda$ should be tuned accordingly.

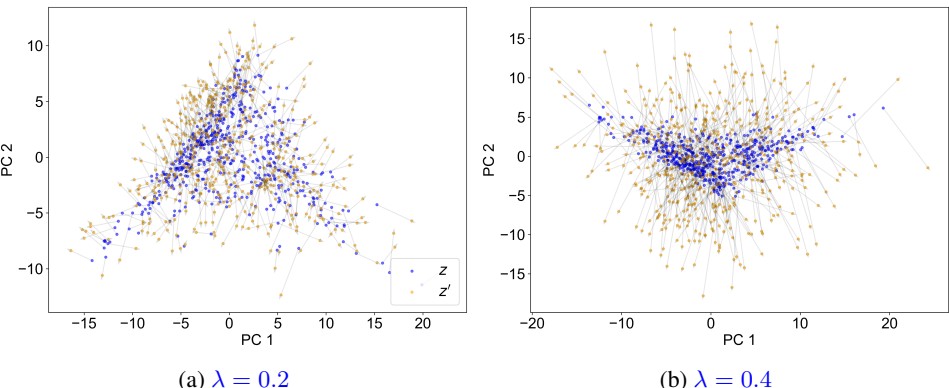

Figure E.3: Principal components (PC) for a batch of data point representations **before** ($z$) and **after** ($z'$) concept-based interventions under the varying values of the parameter for (a) $\lambda = 0.2$ and (b) $\lambda = 0.4$.

### E.3 CUB RESULTS

In line with the previous literature (Koh et al., 2020), we assess our approach on the CUB dataset with the results summarised in Figure E.4. This dataset is similar to the AwA2, as the concepts are shared across whole classes. Thus, concepts and classes feature a strong and simple correlation structure. Expectedly, the CBM performs very well due to its inductive bias in relying on the concept variables. As in the previous simpler scenarios, untuned black boxes are, in principle, intervenable. However, the proposed fine-tuning strategy considerably improves the effectiveness of interventions. On this dataset, the performance (without interventions) of the post hoc CBM and the model fine-tuned for intervenability is visibly lower than that of the untuned black box. We attribute this to the poor association between the concepts and the representations learnt by the black box. Interestingly, post hoc CBMs do not perform as successfully as the models fine-tuned for intervenability. Generally, the behaviour of the models on this dataset falls in line with the findings described in the main body of the text and supports our conclusions.

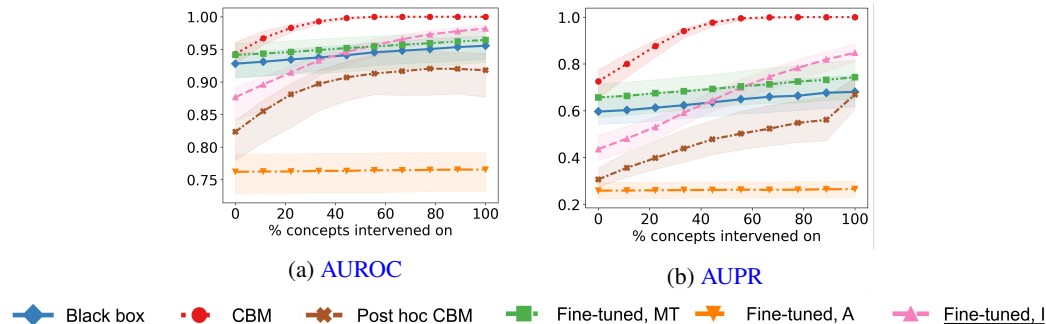

(a) AUROC   (b) AUPR

Figure E.4: Intervention results w.r.t. target AUROC (a) and AUPR (b) across ten initialisations on the CUB dataset.

## E.4    CALIBRATION RESULTS

The fine-tuning approach introduced leads to better-calibrated predictions (Table 1), possibly, due to the regularising effect of intervenability. In this section, we further support this finding by visualising calibration curves for the binary classification tasks, namely, for the synthetic tabular data and chest radiograph datasets. Figure E.5 shows calibration curves for the fine-tuned model, untuned black box, and CBM averaged across ten seeds. We have omitted fine-tuning baselines in the interest of legibility since their predictions were comparably ill-calibrated as for the black box. The fine-tuned model consistently and considerably outperforms both the untuned black box and the CBM in all three binary classification tasks, as its curve is the closest to the diagonal, which corresponds to perfect calibration.

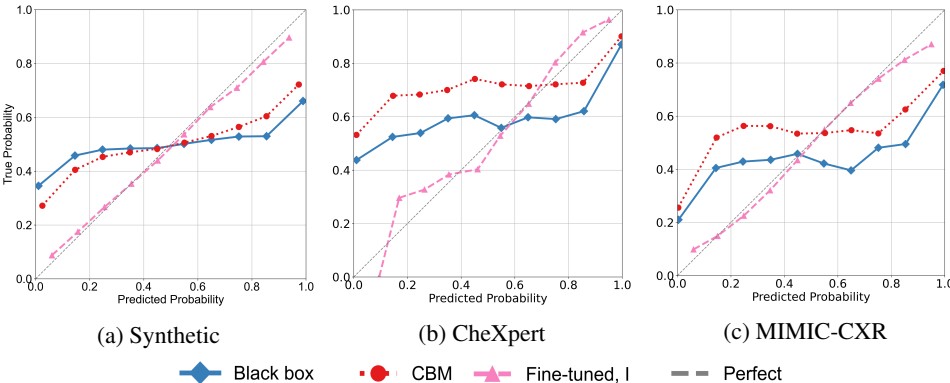

      (a) Synthetic            (b) CheXpert           (c) MIMIC-CXR

Figure E.5: Analysis of the probabilities predicted by the **black box**, **fine-tuned black box**, and **CBM** on the (a) synthetic, (b) CheXpert, and (c) MIMIC-CXR. The calibration curves, averaged across ten seeds, display for each bin the true empirical probability of $y = 1$ against the probability predicted by the model. The gray dashed line corresponds to perfectly calibrated predictions.

## E.5 INFLUENCE OF THE DISTANCE FUNCTION

Throughout the experiments, we have consistently utilised the Euclidean distance as $d$ in Equation 1. In this section, we explore the influence of this design choice. In particular, we fine-tune the black-box model and intervene on all models under the cosine distance given by $d(\boldsymbol{x}, \boldsymbol{x}') = 1 - \boldsymbol{x} \cdot \boldsymbol{x}' / (\|\boldsymbol{x}\|_2 \|\boldsymbol{x}'\|_2)$.

Figure E.6 shows the intervention results under the cosine distance on the four datasets considered before. Firstly, for the synthetic and AwA2 datasets, we observe that the untuned black box is visibly less intervenable than under the Euclidean distance (*cf.* Figures 3 and 4). In fact, for the AwA2 (Figure E.6(b), *top*), interventions slightly reduce the test-set AUROC. These results suggest that the intervention procedure is, indeed, sensitive to the choice of the distance function, and we hypothesise that the distance should be chosen to suit the latent space of the neural network considered. Encouragingly, the proposed fine-tuning procedure is equally effective under the cosine distance. Similar to the Euclidean case, it greatly improves the model's intervenability at test time and bridges the gap towards CBMs.

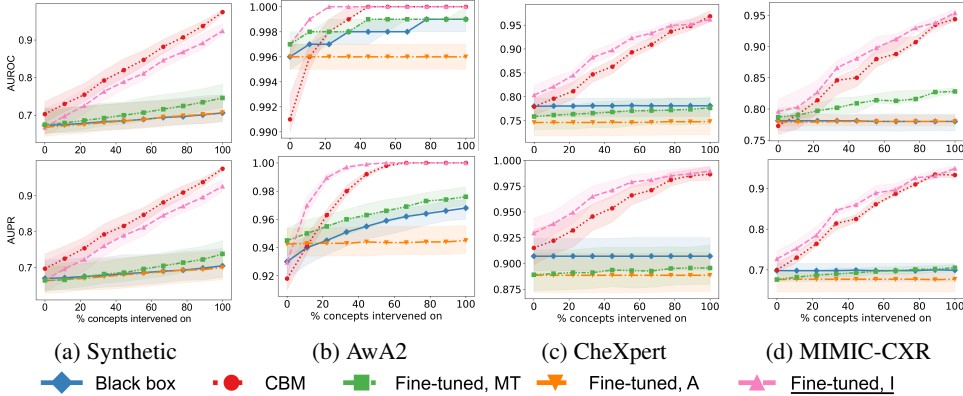

(a) Synthetic  (b) AwA2  (c) CheXpert  (d) MIMIC-CXR

Black box ・ CBM ・ Fine-tuned, MT ・ Fine-tuned, A ・ Fine-tuned, I

Figure E.6: Intervention results w.r.t. target AUROC (*top*) and AUPR (*bottom*) under the cosine distance function (Equation 1) on the four studied datasets: (a) synthetic, (b) AwA2, (c) CheXpert, and (d) MIMIC-CXR. The comparison is performed across ten seeds.

