# OpenReview forum: "Beyond Concept Bottleneck Models: How to Make Black Boxes Intervenable?"
_ICLR.cc/2024/Conference — Submitted to ICLR 2024_

### Official Review · Reviewer_Emzi · 2023-10-22

**Soundness:** 3 good
**Presentation:** 2 fair
**Contribution:** 2 fair
**Rating:** 5
**Confidence:** 3

**Summary:**

This paper proposes a method to make conceptual interventions by modifying the activations of a pre-trained black-box model. Based on the popular counterfactual optimization approach by Wachter et al [1] for pixel-based counterfactuals, they propose to instead optimize for activations. That is, they apply the distance loss in the activation instead of pixel space and use the gradients obtained by an additionally trained concept detector ((non)-linear probe) instead of the gradients of the classifier (c.f., Eq. 1). Finally, the authors propose a fine-tuning scheme to make the classifier more reliant on the “concept activation vectors” based on a proposed notion of intervenability (Eq. 2 & 3), while keeping the feature extractor frozen. Through experiments on tabular and vision data the authors show the efficacy of their intervention as well as fine-tuning scheme.

[1] Wachter, Sandra, et al. "Counterfactual explanations without opening the black box: Automated decisions and the GDPR." Harv. JL & Tech. 31 2017: 841.

**Strengths:**

* The work addresses an interesting problem to better understand the behavior of models through conceptual interventions.
* The intervention strategy is simple yet effective.
* The fine-tuning scheme is well-designed and yet simple.
* Code is provided via an anonymized repository and supplementary material.

**Weaknesses:**

* The work seems to have missed the most relevant work on conceptual (interventional) counterfactuals, e.g., [1,2]. While there are some technical differences (usage of the gradients stemming from the linear probe instead of the classifier in the counterfactual optimization problem), there is still significant overlap. For example, Abid et al. [1] also use linear probes to identify concept activation vectors and use it to intervene on the features.

* The work motivates their approach by stating that “concept labels are not required [during training]” (p. 2). While true, it is still required for the fine-tuning (as we need to train the probes initially; Sec. 3.3, and shown to be important in the experiments), which one could also see as part of model development. For the case that we would like to make the same conceptual interventions as for CBMs, this would result in a similar amount of annotation cost.

* The paper makes the (implicit) assumption that the feature extractor $h_{\phi}$ learns the concept; both in their intervention approach and fine-tuning (since $\beta$ is set to 1). However, the authors provide no evidence that this is actually the case. The linear probes could just learn to predict from some correlated concept/feature. Further evidence would be required that the black-box feature extractor has actually learned the concept that is intervened on.

* It is very unclear whether the intervention strategy actually results in *plausible* and not just *adversarial* changes of the activations. This is a prominent problem for pixel-spaced counterfactuals methods, where, e.g., different types of regularization or generative models are used to obtain plausible counterfactuals.

* There are no comparisons to prior work that converted pretrained models into CBMs [3,4]. It would be interesting to show how the proposed method compares to them (using the proposed intervention strategy).

[1] Abid, Abubakar et al. "Meaningfully explaining model mistakes using conceptual counterfactuals." ICML 2022.

[2] Kim, Siwon, et al. "Grounding Counterfactual Explanation of Image Classifiers to Textual Concept Space." CVPR 2023.

[3] Yuksekgonul, Mert et al. "Post-hoc concept bottleneck models." ICLR 2023.

[4] Oikarinen, Tuomas, et al. "Label-Free Concept Bottleneck Models." ICLR 2023.

**Questions:**

* Why does Eq. 2 & 3 assume that c’ does not change y to y’? As is, it assumes that the concept does not change the class. Let’s say the concept c’ changes the fur texture of a cat, then the resulting class may also change. This seems not to be included in the current notion of intervenability and may be easily obtained by a suitable choice of the distribution $\pi(c’,y’|x,\hat{c},c,\hat{y},y)$, which subsumes $\pi(c’|x,\hat{c},c,\hat{y},y)$ when $y’=y$.

* Why is the ResNet-18 architecture used with four fully-connected layers instead of the standard setting? Why are not just the bottleneck features used?

* What happens for the baseline (fine-tuned, MT) if we interleave intervened activations $z’$ also during training? As is, it just may be that (fine-tuned, I) is more robust to the interventions.

* Are AUROC and AUPR computed for the concepts or targets/classes? This may also change for the different experimental results (e.g., for the figures in the main paper this is unclear). Could the authors clarify this?

## Suggestions

* It’d have been good to discuss the data-generating mechanisms (bottleneck, confounder, incomplete) in the main text and not only supplemental.

* Given the overlapping confidence bands in Fig. 3(c) it would be good to either run more simulations or reformulate the sentence since it is unclear if “black-box classifiers are expectedly superior to the CBM”.

---

> ### Author Response · Authors · 2023-11-17
> **Point-by-point Response to Reviewer Emzi (Part 1)**
>
> We thank the reviewer for the thorough feedback! Below are our point-by-point responses to the concerns raised.
>
> > The work seems to have missed the most relevant work on conceptual (interventional) counterfactuals, e.g., [1,2]. While there are some technical differences (usage of the gradients stemming from the linear probe instead of the classifier in the counterfactual optimization problem), there is still significant overlap. For example, Abid et al. [1] also use linear probes to identify concept activation vectors and use it to intervene on the features.
>
> We thank the reviewer for pointing us to this line of work! In the revised manuscript, we have included references to [[1]](https://proceedings.mlr.press/v162/abid22a.html) and [[2]](https://openaccess.thecvf.com/content/CVPR2023/html/Kim_Grounding_Counterfactual_Explanation_of_Image_Classifiers_to_Textual_Concept_Space_CVPR_2023_paper.html). We have also provided a detailed discussion of the differences between ours and these related works in Appendix B. Below, we provide a general summary.
>
> While some of the technical tricks used by our work and conceptual counterfactual explanations (CCE) [1, 2] are similar, the problem setting and the actual purpose are quite different. Informally, our intervention addresses the following question: “*How do the network’s representations need to be perturbed to reflect the user-input concepts $\boldsymbol{c}’$?*” On the other hand, CCEs try to tackle the following question: “*Which concepts need to be input for perturbing the network’s representations to flip the predicted label $\hat{y}$ to the given $y’$?*”. In summary, CCEs try to identify concept variables responsible for a misclassification, whereas interventions try to improve a prediction at test time based on the input concepts.
>
> In our method, interventions are intended for the interaction between the user and the model so that the user can reduce the model’s error via understandable concept variables. By contrast, CCEs consider a typical counterfactual explanation scenario, trying to identify which concepts can lead to sparse and in-distribution changes in the representation that would flip the classification outcome.
>
> Beyond the main difference above, our work makes a few other technically valuable contributions. Namely, we formalise the intervenability measure and intervention strategies and propose explicit fine-tuning for intervenability. In summary, our work tackles a different problem complementary to the direction explored by [1] or [2].
>
> > The work motivates their approach by stating that “concept labels are not required [during training]” (p. 2). While true, it is still required for the fine-tuning (as we need to train the probes initially; Sec. 3.3, and shown to be important in the experiments), which one could also see as part of model development. For the case that we would like to make the same conceptual interventions as for CBMs, this would result in a similar amount of annotation cost.
>
> Fine-tuning indeed requires concept labels. We would like to emphasise that all the experiments were performed on *validation* sets, which were considerably smaller than the training sets (utilised by CBMs). Moreover, labelling costs are not the only concern when using CBMs. In particular, concept variables might be unknown when training a model and only be incorporated when deploying the model for a specific application in some institution (e.g. hospital). For instance, such challenges may arise when utilising foundation models [[5]](https://doi.org/10.1038/s41586-023-05881-4).
>
> > The paper makes the (implicit) assumption that the feature extractor $h_{\phi}$ learns the concept; both in their intervention approach and fine-tuning (since $\beta$ is set to 1). However, the authors provide no evidence that this is actually the case. The linear probes could just learn to predict from some correlated concept/feature. Further evidence would be required that the black-box feature extractor has actually learned the concept that is intervened on.
>
> For interventions and fine-tuning to be effective, the assumption outlined by the reviewer needs to hold. This assumption was explicitly mentioned in the paragraph “Should All Models Be Intervenable?” in Section 3.2 of the initially submitted manuscript (this paragraph has been moved to Appendix B in the revised version).
>
> In our view, if the interventions improve the predictive performance (before or after fine-tuning), this implies that the representations $\boldsymbol{z}$ are informative of the concept variables and, hence, interventions can be performed somewhat effectively. We do not deem it necessary that representations *identify* concept variables or are *disentangled* w.r.t. the concepts.

---

> > ### Author Response · Authors · 2023-11-17
> > **Point-by-point Response to Reviewer Emzi (Part 2)**
> >
> > > It is very unclear whether the intervention strategy actually results in plausible and not just adversarial changes of the activations. This is a prominent problem for pixel-spaced counterfactuals methods, where, e.g., different types of regularization or generative models are used to obtain plausible counterfactuals.
> >
> > Along the lines of the discussion above, our interventions aim to inject given concept information and update the model’s output to reduce the error. Hence, producing plausible changes in the representations seemingly is not a concern for the current method and would rather be relevant from the perspective of counterfactual explanation [1].
> >
> > The degree of perturbation can be controlled via the parameter $\lambda$ in Equation 1. In particular, higher values of $\lambda$ result in intervened representations more consistent with the given concepts $\boldsymbol{c}’$ but more distant from the original representations. We provide a brief exploration of this in Appendix E (Figure E.3) of the revised manuscript.
> >
> > > There are no comparisons to prior work that converted pretrained models into CBMs [3,4]. It would be interesting to show how the proposed method compares to them (using the proposed intervention strategy).
> >
> > As suggested by the reviewer, we have included a comparison with post hoc CBMs [[3](https://arxiv.org/abs/2205.15480), [4](https://arxiv.org/abs/2304.06129)] in the revised manuscript. We have observed that post hoc CBMs perform quite well on simpler datasets but are somewhat less intervenable than fine-tuned models.
> > By contrast, for the synthetic incomplete and chest X-rays, interventions on post hoc CBMs are either ineffective or harmful. We attribute this to potential leakage [[6]](https://proceedings.neurips.cc/paper_files/paper/2022/hash/944ecf65a46feb578a43abfd5cddd960-Abstract-Conference.html). In general, these findings further confirm our claim that explicitly fine-tuning for intervenability is crucial for making interventions on black-box models effective.
> >
> > > Why does Eq. 2 & 3 assume that c’ does not change y to y’? As is, it assumes that the concept does not change the class. Let’s say the concept c’ changes the fur texture of a cat, then the resulting class may also change. This seems not to be included in the current notion of intervenability and may be easily obtained by a suitable choice of the distribution.
> >
> > We do not assume that the intervention has to change the prediction to a specific $y’$. The intervened concepts $\boldsymbol{c}’$ are meant to be provided by the user, and the strategy $\pi$ represents this process of querying the expert. The goal of interventions is not the counterfactual explanation, i.e. flipping the label, as in [1], but rather the injection of expert-provided knowledge of the concept variables.
> >
> > Moreover, $y$ in Equations 2 and 3 corresponds to the ground-truth label, i.e. intervenability measures the gap in the loss without and with interventions, which are meant to steer the model towards predicting the correct label, reducing its error and improving calibration. In general, however, interventions need not change the model’s prediction if the representations are consistent with the given concepts and the initial prediction is correct.

---

> > > ### Author Response · Authors · 2023-11-17
> > > **Point-by-point Response to Reviewer Emzi (Part 3)**
> > >
> > > > Why is the ResNet-18 architecture used with four fully-connected layers instead of the standard setting? Why are not just the bottleneck features used?
> > >
> > > We chose to have a few fully connected layers after the ResNet-18 backbone to demonstrate that interventions could be performed at an intermediate layer and impact a nonlinear link function. We expect comparable (or even better) results for intervening in the standard setting mentioned by the reviewer.
> > >
> > > > What happens for the baseline (fine-tuned, MT) if we interleave intervened activations $z’$ also during training? As is, it just may be that (fine-tuned, I) is more robust to the interventions.
> > >
> > > If we were to interleave intervened representations $\boldsymbol{z}’$ when training the Fine-tuned MT baseline, this trick would make the baseline very similar to fine-tuning for intervenability (Fine-tuned, I) for those data points where the original representation had been replaced with $\boldsymbol{z}’$. Hence, this would not be a very fair baseline.
> > >
> > > > Are AUROC and AUPR computed for the concepts or targets/classes? This may also change for the different experimental results (e.g., for the figures in the main paper this is unclear). Could the authors clarify this?
> > >
> > > Figures 3, 4, and 5 depict changes in the *target* AUROC and AUPR under interventions. Table 1 shows concept and target prediction performance metrics (AUROC, AUPR, and Brier score) *without* interventions. We have adjusted figure and table captions to make them clearer in the revised version of the manuscript.
> > >
> > > > It’d have been good to discuss the data-generating mechanisms (bottleneck, confounder, incomplete) in the main text and not only supplemental.
> > >
> > > We have included a brief description of these generative mechanisms in the main text of the revised manuscript.
> > >
> > > > Given the overlapping confidence bands in Fig. 3(c) it would be good to either run more simulations or reformulate the sentence since it is unclear if “black-box classifiers are expectedly superior to the CBM”.
> > >
> > > We would like to note that the bands shown in the figures are interquartile ranges over ten independent *simulations* (rather than only initialisations or train-validation-test splits). Confidence intervals for the mean are considerably narrower. For improved replicability, we will re-run this experiment for a larger number of replicates when preparing the next iteration of the manuscript.
> > >
> > > ### References
> > >
> > > [1] Abid, A., Yuksekgonul, M. & Zou, J.. (2022). Meaningfully debugging model mistakes using conceptual counterfactual explanations. *Proceedings of the 39th International Conference on Machine Learning*, in *Proceedings of Machine Learning Research* 162:66-88.
> > >
> > > [2] Kim, S., Oh, J., Lee, S., Yu, S., Do, J., & Taghavi, T. (2023). Grounding Counterfactual Explanation of Image Classifiers to Textual Concept Space. In *Proceedings of the IEEE/CVF Conference on Computer Vision and Pattern Recognition* (pp. 10942-10950).
> > >
> > > [3] Yuksekgonul, M., Wang, M., & Zou, J. (2022). Post-hoc concept bottleneck models.
> > > *arXiv:2205.15480*.
> > >
> > > [4] Oikarinen, T., Das, S., Nguyen, L. M., & Weng, T. W. (2023). Label-Free Concept Bottleneck Models. *arXiv:2304.06129*.
> > >
> > > [5] Moor, M., Banerjee, O., Abad, Z. S. H., Krumholz, H. M., Leskovec, J., Topol, E. J., & Rajpurkar, P. (2023). Foundation models for generalist medical artificial intelligence. *Nature, 616*(7956), 259-265.

---

> > > > ### Comment · Reviewer_Emzi · 2023-11-21
> > > > **Re: Point-by-point Response to Reviewer Emzi**
> > > >
> > > > I thank the authors for the thorough reply. I acknowledge the different problem settings of conceptual counterfactuals and the present work’s setting, and appreciate the discussion in Appendix B.
> > > > > Fine-tuning indeed requires concept labels.
> > > >
> > > > It’d be good if the authors could more clearly state this in their work and rephrase sentences such as “our method circumvents the need for concept labels during training” (p. 8).
> > > >
> > > > > For interventions and fine-tuning to be effective, the assumption [that the feature extractor has already learned the concept, …] needs to hold.
> > > >
> > > > I believe this is an important question that requires further experimental evaluation, as the goal is to obtain an interpretable model in the sense that intervened concepts have the intended behavior, as also thoroughly discussed in the paragraph “Should All Models Be Intervenable?” in Appendix B. The need for fine-tuning may hint that this is not the case.
> > > >
> > > > > We do not deem it necessary that representations identify concept variables or are disentangled w.r.t. the concepts.
> > > >
> > > > Could the authors extend what they mean with this?
> > > >
> > > > > If we were to interleave intervened representations when training the Fine-tuned MT baseline, this trick would make the baseline very similar to fine-tuning for intervenability
> > > >
> > > > $z’$ may be out-of-distribution for $g_\psi$ and could explain its worse performance. Thus, it’d be interesting to consider this as a baseline.

---

> > > > > ### Author Response · Authors · 2023-11-21
> > > > > **Follow-up to Reviewer Emzi**
> > > > >
> > > > > Thank you very much for the prompt follow-up!
> > > > >
> > > > > > It’d be good if the authors could more clearly state this in their work and rephrase sentences such as “our method circumvents the need for concept labels during training” (p. 8).
> > > > >
> > > > > In case of acceptance, we will adjust this passage in the *Discussion* as follows: “*In contrast to interpretable models such as CBMs (Koh et al., 2020), our method circumvents the need for concept labels during training, which can be a substantial challenge in practical applications. Instead, our method relies on a smaller annotated validation set to train probing functions and perform fine-tuning.*” We will also make similar adjustments in other passages where applicable, e.g. on p. 2 in the *Contributions* paragraph.
> > > > >
> > > > > > I believe this is an important question that requires further experimental evaluation, as the goal is to obtain an interpretable model in the sense that intervened concepts have the intended behavior, as also thoroughly discussed in the paragraph “Should All Models Be Intervenable?” in Appendix B. The need for fine-tuning may hint that this is not the case.
> > > > >
> > > > > Indeed, the feature extractor should learn representations that are sufficiently correlated with the concept information for the model to be intervenable. The need for fine-tuning indicates that not all black-box models utilise concept information downstream. This happens despite having representations that might be correlated with the concepts (as indicated by the concept prediction results without interventions), e.g. random projections would also feature a strong correlation with concept variables. Our experimental results illustrate this quite well: in some cases, the heads of black-box models do not rely on concept variables; after fine-tuning, they are recalibrated accordingly. Finally, note that the assumption in question is also standard across related works, such as [1] and [2], which likewise utilise the frozen backbone, assuming representations to be predictive of concepts.
> > > > >
> > > > > Furthermore, note that the proposed method is not positioned as seeking inherent interpretability. As discussed in the paragraph “*Should All Models Be Intervenable?*” in Appendix B, if the data contain correlations among features, concepts, and targets, we want to steer the model through the concept information to influence the target prediction. Generally, the layer we intervene on in black box models does not have to be directly interpretable, as is usually the case in complex datasets and models, mainly because no concept knowledge was given at the training time. Hereafter, fine-tuning with concept information is necessary to enforce intervenability, not by altering the representations, but the downstream predictor $g_{\boldsymbol{\psi}}$. Our results show that fine-tuning improves model intervenability since we observe increased performance with interventions across all datasets. Of course, this is feasible because, in all cases, networks’ representations are initially correlated with concepts.
> > > > >
> > > > > > Could the authors extend what they mean with this?
> > > > >
> > > > > By not having to identify concept variables or being disentangled w.r.t. the concepts, we mean that *individual* representation dimensions do not have to be *uniquely* associated with specific concept variables. In other words, for performing interventions, it suffices that $\boldsymbol{z}$ are merely correlated with $\boldsymbol{c}$ so that a probe can be trained and consequently $\boldsymbol{z}’$ could be derived.
> > > > >
> > > > > > $z’$ may be out-of-distribution for $g_{\psi}$ and could explain its worse performance. Thus, it’d be interesting to consider this as a baseline.
> > > > >
> > > > > This is indeed one of the possible reasons why Fine-tuned MT performs worse than Fine-tuned I. However, please note that if Fine-tuned MT were to be trained on $\boldsymbol{z}’$, it would effectively maximise intervenability on a fraction of data points where $\boldsymbol{z}’$ was used. Hence, it would be impossible to disentangle the effect of the proposed intervenability loss from multitask learning for such a baseline. Therefore, such a baseline would effectively correspond to Fine-tuned I and, in our view, would not be fair or very meaningful. As an analogy, think of comparing a CBM to a black-box model where dimensions of some layer in some data points are explicitly supervised with the concept prediction loss (the latter method is essentially a CBM and would not constitute a very meaningful baseline).
> > > > >
> > > > > ### References
> > > > >
> > > > > [1] Yuksekgonul, M., Wang, M., & Zou, J. (2022). Post-hoc concept bottleneck models.
> > > > > *arXiv:2205.15480*.
> > > > >
> > > > > [2] Oikarinen, T., Das, S., Nguyen, L. M., & Weng, T. W. (2023). Label-Free Concept Bottleneck Models. *arXiv:2304.06129*.

---

> > > > > > ### Comment · Reviewer_Emzi · 2023-11-22
> > > > > > **Re: Follow-up to Reviewer Emzi**
> > > > > >
> > > > > > I thank the authors again for their reply. I acknowledge the authors’ effort to revise the manuscript and provide further clarifications. However, some of my concerns remain, including the assumption of a concept-extracting feature encoder and uncertainty whether the activation vector $z’$ truly results in the intended intervention, considering that it could be just an out-of-distribution activation for both the probe $q_\xi$ and classifier head $g_\psi$, as evidenced by the need for fine-tuning. Nonetheless, I will change my overall score to 5.

---

> > > > > > > ### Author Response · Authors · 2023-11-22
> > > > > > > **Thank you!**
> > > > > > >
> > > > > > > Thank you very much for engaging in the discussion and adjusting your score!
> > > > > > >
> > > > > > > We would like to stress that the distribution of intervened representations $\boldsymbol{z}'$ is explored in Appendix E.2 of the revised manuscript: as we observe, hyperparameter $\lambda$ has an effect on the difference between original representations $\boldsymbol{z}$ and $\boldsymbol{z}'$. In our view, being out-of-distribution is not very consequential for the application scenario we study. We agree with the reviewer that this could be one of the reasons why fine-tuning helps. However, we do not see this as a limitation of our work but rather an additional motivation for the introduced fine-tuning technique.

---

> ### Author Response · Authors · 2023-11-21
> **Follow-up on the Rebuttal**
>
> Dear reviewer Emzi,
>
> Given that the end of the discussion period is approaching, we would like to ask if you have any further concerns or questions, particularly as a follow-up to our response? Thank you in advance!

---

### Official Review · Reviewer_icNv · 2023-10-25

**Soundness:** 3 good
**Presentation:** 3 good
**Contribution:** 3 good
**Rating:** 8
**Confidence:** 4

**Summary:**

The authors introduce a method to perform concept-based interventions on pre-trained neural networks. They then formalize intervenability as a metric to measure concept-based interventions. Finally, they show that finetuning probes for intervenability can improve intervenability.

**Strengths:**

- The paper's proposed metric of intervenability is interesting and helps practically measure the utility of methods such as CBMs
- The introduced methods are clear and intuitive
- Strong comparisons are made to baselines from previous work

**Weaknesses:**

- Limited improvements over CBMs -- In most settings, CBMs seem to outperform the proposed Fine-tuned, I method while providing much greater interpretability (the main exception is in Fig 5)
- Potential missing baseline: I am not sure if a simple convex combination of the predictions of the CBM and the black-box would outperform the proposed model

**Questions:**

- Note: would be nice for the paper to mention the link between intervenability and concept/feature importance - this should be straightforward based on interventions, as this is how many feature importance metrics (e.g. LIME/SHAP) are computed
- Would be nice to describe the experimental setup in more detail, e.g. the three synthetic scenarios are only described in previous work
- Minor: fig legends in Fig 4 are slightly difficult to read.

---

> ### Author Response · Authors · 2023-11-17
> **Point-by-point Response to Reviewer icNv**
>
> We thank the reviewer for the feedback! Below are our point-by-point responses to the concerns raised.
>
> > Limited improvements over CBMs -- In most settings, CBMs seem to outperform the proposed Fine-tuned, I method while providing much greater interpretability (the main exception is in Fig 5).
>
> CBMs are indeed more intervenable than black-box models (even after fine-tuning) on *simple* benchmarks, such as synthetic bottleneck and confounder (where generative mechanism directly or closely matches the CBM). However, on the synthetic incomplete and chest X-ray datasets, where the set of concepts likely does not fully explain the relationship between $\boldsymbol{x}$ and $y$, fine-tuned black boxes outperform CBMs. These results are not surprising, and we would argue that the proposed approach can be practical and useful for practitioners working with more complex datasets.
>
> Finally, there exist scenarios where training a CBM is impossible or impractical, e.g. when relying on a foundation model [[1]](https://doi.org/10.1038/s41586-023-05881-4) or when the concept variables are unknown or too expensive to label at the training time. Thus, we believe there is practical merit in the proposed techniques.
>
> > Potential missing baseline: I am not sure if a simple convex combination of the predictions of the CBM and the black-box would outperform the proposed model.
>
> While the suggested baseline might be practical, combining a CBM with a black-box model would lead to interpretations and interventions that are not faithful to the predictions made since the output of the CBM would be augmented with the black-box prediction that is not intervenable or explained in any way. In addition, as mentioned above, our work is motivated by application scenarios in which CBMs are not as practical.
>
> > Note: would be nice for the paper to mention the link between intervenability and concept/feature importance - this should be straightforward based on interventions, as this is how many feature importance metrics (e.g. LIME/SHAP) are computed.
>
> Thank you for this suggestion! There indeed exists a relationship between intervenability and feature importance measures.
>
> Drawing a direct relationship with LIME or SHAP is somewhat challenging since these feature importance measures assess the expected change in the model’s output rather than loss. On the other hand, intervenability is very reminiscent of the model reliance as formalised in [[2]](https://arxiv.org/abs/1801.01489) and explained informally in [[3]](https://christophm.github.io/interpretable-ml-book/feature-importance.html#theory-3), which is another family of feature importance measures. In particular, model reliance ($MR$) on feature $j$ is given by the ratio of the expected loss of the model $f$ when the feature $j$ is augmented with noise, rendering this feature uninformative of the target variable versus the expected loss under the original data distribution. Similarly, the intervenability (as defined in Equation 2) measures the *difference* in the expected losses. Suppose the distribution $\pi$ (intervention strategy) is chosen to augment a single concept variable with the noise (as described in [[2]](https://arxiv.org/abs/1801.01489)). In that case, intervenability may be used to quantify the reliance of $g_{\boldsymbol{\psi}}$ on the individual concepts in $\boldsymbol{\hat{c}}$. We have added a comment on these observations in the revised manuscript in Appendix B.
>
> > Would be nice to describe the experimental setup in more detail, e.g. the three synthetic scenarios are only described in previous work.
>
> Note that the synthetic dataset and three different generative mechanisms were described in detail in the appendix of the submitted manuscript. To improve clarity, we have added a brief description of the three scenarios to the main text of the revised manuscript. We have also included a detailed explanation of the intervention strategies in Appendix D.
>
> ### References
>
> [1] Moor, M., Banerjee, O., Abad, Z. S. H., Krumholz, H. M., Leskovec, J., Topol, E. J., & Rajpurkar, P. (2023). Foundation models for generalist medical artificial intelligence. *Nature, 616*(7956), 259-265.
>
> [2]  Fisher, A., Rudin, C., & Dominici, F. (2019). All Models are Wrong, but Many are Useful: Learning a Variable's Importance by Studying an Entire Class of Prediction Models Simultaneously. *Journal of Machine Learning Research, 20*(177), 1-81.
>
> [3] Molnar, C. (2020). *Interpretable machine learning*. Lulu. com.

---

> ### Comment · Reviewer_icNv · 2023-11-21
> **Raised my score to 8**
>
> I have raised my score to 8 in response to the author's comments. It would still be nice to concretely see scenarios where CBMs fail and the proposed method shows a significant improvement, but the authors have done a decent job supporting that these scenarios exist.

---

> > ### Author Response · Authors · 2023-11-21
> > **Thank you!**
> >
> > Thank you for the follow-up and updated score! We appreciate the discussion, your feedback and suggestions.

---

### Official Review · Reviewer_39mA · 2023-10-28

**Soundness:** 3 good
**Presentation:** 3 good
**Contribution:** 4 excellent
**Rating:** 8
**Confidence:** 4

**Summary:**

- The paper proposes interventions on black-box models in CBM style.


- Given a black-box model:
     -  Train a probing network to extract concepts from an intermediate representation $z$
     -  Now that they have the probing network for a given sample $x$ they can extract $z$ and concepts $c$, given the ground-truth concepts $c'$ they learn a new embedding $z'$ that should produce concepts $c'$ when given to probing network.
     -  For interventions they replace the original $z$ with new $z'$

- Given $z$ how do we calculate $z'$? (from the text is a bit unclear I had to look at the code to understand how this is done so please correct me if I am wrong)
     -    Start from $z'==z$ but $z'$ is differentiable.
     -    The probing network is frozen where you calculate concepts $q_\xi(z')$
     -    Update $z'$ based on the loss in equation 1 you repeat multiple times i.e for a few epochs.


- The paper quantifies the effectiveness of interventions as the gap between the regular prediction loss and the loss attained after the intervention.
- The paper proposes a fine-tuning strategy for intervention that can be summarized as follows:
     -    Given a black-box model $f_\theta$, we will look at the network as if its a cbm model such that we first have a network $h_\phi(x)=z$ that gives us the intermediate representation used to train the probing network and $g_\psi(z)=y$ that gives the final prediction.
     -    The black box network is now trained end to end using loss in equation 4: the first term is regular black-box optimization the second term is optimizing the outer network subjected to the intervention and all of this is in addition to optimizing $z'$  as mentioned previously.
     -    Equation 4 is then simplified to avoid trilevel optimizing the first loss is ignored so the feature extractor layer is basically frozen.
-  Experiments:
    -    The paper one synthetic tabular dataset, 3 image datasets.
    -    The paper tested the proposed fine-tuning approach against a black-box NN a CBM and two other different fine-tuning approaches.

**Strengths:**

Strength:
- Originality: The paper is original, there has been work around post-hoc CBM (cited by the paper) but this intervention strategy is novel and very interesting.
- Quality: The paper is evaluated against a reasonable baseline on multiple datasets.
- Significance: The paper's contribution is significant, if we can intervene on model in test time we can get higher accuracy as shown on multiple datasets.

**Weaknesses:**

Method weakness:
- There is no guarantee that a network has learned the desired concepts, i.e. there is a big probability that the probing network can not learn a concept you would want to intervene on.
- The method is quite expensive optimizing to get the $z'$ and optimizing for fine-tuning on top of $z'$ can be quite costly.
- Creating a probing network per concept can be costly when we have a large number of concepts, it is not clear if this can scale to hundreds or thousands of concepts.
- It is not clear which concepts one should intervene on.
- What if we can never intervene during test time (say we don't have ground-truth concepts or even ground-truth label at that point) it is unclear how this can be useful in that case.


Paper quality:
- How you get $z'$ and the intervening strategy is not very clear in the main paper I would strongly recommend moving Algorithm A.1 to the main text for clarity.

**Questions:**

- Which layer do you do probing on is it the last layer before the classifier?
- Usually how many iterations do you need to extract a reasonable $z'$.
- In the experiments, the accuracy on the test set is "with" interventions correct?
- How do you select the concepts to intervene on?
- How would this model be used practically? (I am assuming something similar to the following steps):


    - You have an example that is incorrect
    - You calculate the concepts that affect that example.
   - You show the concepts to a domain expert, and they propose different concepts.
   - You calculate $z'$ using this new concept and make a new prediction.

If so how is the original model improved it?

---

> ### Author Response · Authors · 2023-11-17
> **Point-by-point Response to Reviewer 39mA (Part 1)**
>
> We thank the reviewer for the positive evaluation of our work and detailed feedback! Below are our point-by-point responses to the concerns raised.
>
> > The paper proposes interventions on black-box models in CBM style.
>
> Thank you for the neat and accurate summary of our work! We appreciate your effort.
>
> > There is no guarantee that a network has learned the desired concepts, i.e. there is a big probability that the probing network can not learn a concept you would want to intervene on.
>
> This limitation is indeed one of the caveats of the proposed intervention recipe. Nevertheless, in the scenario described by the reviewer, the intervenability measure alongside fine-tuning can help practitioners understand that the black-box model is perhaps inadequate for the considered application. In that case, we should resort to (i) in addition, fine-tuning w.r.t. the parameters of the feature extractor $h_{\boldsymbol{\phi}}$ or (ii) using another black-box model.
>
> > The method is quite expensive optimizing to get the $z’$ and optimizing for fine-tuning on top of $z’$  can be quite costly.
>
> Interventions require additional optimisation; however, they can be performed on a batch of data points to speed up the inference. By contrast, the fine-tuning is run only once on the validation set (smaller than the training set), and at deployment, only the intervention routine needs to be executed.
>
> > Creating a probing network per concept can be costly when we have a large number of concepts, it is not clear if this can scale to hundreds or thousands of concepts.
>
> We would like to emphasise that we use a *multivariate* probing function in all experiments (for both linear and nonlinear functions), i.e. the probe predicts concept variables in a multitask fashion. Thus, we expect no considerable additional scaling issues when dealing with a large number of concept variables. Please note that, for example, the AwA2 and CUB datasets have 85 and 112 concept variables, respectively, and we observed no scalability problems for these benchmarks.  Finally, having thousands of concept variables requires a significant mental effort on the user’s side and, in that setting, the interpretability and utility of a concept-based approach are somewhat limited.
>
> > It is not clear which concepts one should intervene on.
>
> Arguably, the expert user should decide which concepts to intervene on, e.g. consider a medical specialist paired up with an ML model predicting the patient’s diagnosis. We represent this decision by the *intervention strategy* $\pi\left(\boldsymbol{c}' \vert \boldsymbol{x}, \boldsymbol{\hat{c}}, \boldsymbol{c}, \hat{y}, y\right)$. Note that the set of conditioning variables depends on the problem setting. In our experiments, we explore two simple strategies: (i) random-subset, where the set of concept variables is chosen uniformly at random and the values of the chosen variables are replaced with the ground truth, and (ii) uncertainty-based, where the variables are sampled with probabilities proportional to the uncertainty given by $1/\left(\left|\hat{c}_j-0.5\right|+\varepsilon\right)$, where $\varepsilon>0$ is small. Generally, other strategies are viable and can be easily incorporated into our method and experiments, e.g. [[1]](https://arxiv.org/abs/2302.14260) proposes a few plausible options and conducts a thorough empirical evaluation.
>
> To improve clarity, we have now included a pseudocode description of the considered intervention strategies in the appendix of the revised paper (Appendix D, Algorithms D.1 and D.2).
>
> > What if we can never intervene during test time (say we don't have ground-truth concepts or even ground-truth label at that point) it is unclear how this can be useful in that case.
>
> As the intended use case of the proposed method is human interaction with a black-box ML model, neither ground-truth concept nor labels are strictly required since $\boldsymbol{c}’$ is supposed to be provided by the user (these could be either ground-truth, golden-standard, or values reflecting the user’s informed guess). As mentioned, the set of conditioning variables for the strategy $\pi$ may vary depending on the application scenario. In fact, none of the intervention strategies considered in our experiments rely on the ground-truth target label $y$.
>
> On the other hand, if the user cannot provide golden-standard or ground-truth concept values, then a concept-based approach is not helpful for the problem at hand, and the user should resort to a black-box model without interventions.

---

> > ### Author Response · Authors · 2023-11-17
> > **Point-by-point Response to Reviewer 39mA (Part 2)**
> >
> > > How you get $z’$ and the intervening strategy is not very clear in the main paper I would strongly recommend moving Algorithm A.1 to the main text for clarity.
> >
> > We agree that, ideally, Algorithm A.1 should have been included in the main text. However, given the limit of 9 pages (applicable to the revised and camera-ready versions as well), we find it challenging to make space for the pseudocode without hurting the manuscript's clarity in other regards. We will consider including its shortened version in the camera-ready in case of acceptance.
> >
> > >  Which layer do you do probing on is it the last layer before the classifier?
> >
> > For the fully connected network architectures, the probing is performed on the last layer before the classifier, as mentioned in Appendix D. For ResNet-18, probing is done on the features. Of course, any layer could be chosen, and this factor is one of the design choices embedded in our method.
> >
> > > Usually how many iterations do you need to extract a reasonable $z’$.
> >
> > The number of iterations required depends on several factors, e.g. the value of the parameter $\lambda$, the number of concept variables, the dimensionality of $z$, and the number of data points in the batch. Given all these considerations, providing a meaningful answer to this question is quite challenging. In our experience, interventions are considerably less costly than training or fine-tuning and can be performed online on a single GPU.
> >
> > For example, on the synthetic dataset, with the convergence parameter set to $10^{-6}$ (a very conservative value) and for a batch of 512 data points, the intervention procedure requires approx. 500 to 2500 steps (the number of steps varies depending on the number of given concepts and the value of $\lambda$), which amounts to 0.5 to 2 seconds (for the whole batch), for an untuned black-box model. We use smaller batches and more permissive convergence criteria when fine-tuning, allowing a considerable speedup. In addition, note that the run-time of the intervention procedure is not strictly dependent on the black-box model’s architecture (except for the dimensionality of the layer intervened on).
> >
> > > In the experiments, the accuracy on the test set is "with" interventions correct?
> >
> > Figures 3, 4, and 5 provide AUROC and AUPR curves *with* the interventions (on varying numbers of concept variables given by percentages). Table 1 evaluates the concept and target predictive performance *without* the interventions.
> >
> > > How do you select the concepts to intervene on?
> >
> > The concept variables to be intervened on are sampled according to the intervention strategy $\pi$. As mentioned before, we have explored two different intervention strategies: sampling concepts uniformly at random (random-subset) and sampling concepts according to the predictions’ uncertainty (uncertainty-based). To clarify this aspect, we have included a detailed description of these strategies in Appendix D in the revised manuscript.

---

> > > ### Author Response · Authors · 2023-11-17
> > > **Point-by-point Response to Reviewer 39mA (Part 3)**
> > >
> > > > How would this model be used practically?
> > >
> > > The use case scenario outlined by the reviewer is quite close to what we have envisioned. 1) The model makes a prediction $\hat{y}$ for some $\boldsymbol{x}$. 2) The user inspects the prediction $\hat{y}$, features $\boldsymbol{x}$, and the probe’s predicted concepts $\boldsymbol{\hat{c}}$. 3) If the user deems it necessary, they intervene by providing an alternative concept vector $\boldsymbol{c}’$, e.g. this could be an edited version of $\boldsymbol{\hat{c}}$ with a few corrections based on the user’s knowledge. 4) We calculate $\boldsymbol{z}’$ given the $\boldsymbol{c}’$ and compute the updated prediction $\hat{y}’$.
> > >
> > > Note that, in practice, it might be challenging to correct the model’s prediction $\hat{y}$ directly, and it might be more sensible to intervene via concept variables. There are several potential reasons for this: (i) the user might lack the expertise to understand and make a final prediction (e.g. think of a nurse interacting with a predictive model rather than a doctor: the nurse is qualified to report and observe findings, but might not be able to identify the diagnosis); (ii) the black-box model might learn a predictive relationship that relies on some information in $\boldsymbol{x}$ not captured by $\boldsymbol{c}$ and unknown to the expert: in that case, our techniques allow the user to correct the model via the variables they understand ($\boldsymbol{c}$); (iii) the relationship between concept variables and $y$ might be nonlinear and not fully understood by the user, e.g. they might be aware of the relevance of the concept variables and their marginal association with the target, but not the higher-order interactions.
> > >
> > > > If so how is the original model improved it?
> > >
> > > We are not entirely sure what the reviewer means by this question. If our answer below does not address it adequately, we would happily have a follow-up discussion.
> > >
> > > The updated prediction $\hat{y}’$ is based on $\boldsymbol{z}’$, and $\boldsymbol{z}’$ is calculated using the provided concepts $\boldsymbol{c}’$. If $\boldsymbol{c}’$ are close to the ground truth concept values and there is a strong association between $\boldsymbol{c}$ and $y$, we expect $\hat{y}’$ to have a lower error than the original $\hat{y}$.
> > >
> > > ### References
> > >
> > > [1]  Shin, S., Jo, Y., Ahn, S., & Lee, N. (2023). A closer look at the intervention procedure of concept bottleneck models. *arXiv:2302.14260*.

---

> ### Comment · Reviewer_39mA · 2023-11-20
> **Thanks!**
>
> I would like to thank the authors for their detailed responses to my concerns and questions.
> I think this work is very interesting and important, and the paper is well-written.
> My score remains as is.

---

> > ### Author Response · Authors · 2023-11-21
> > **Thank you!**
> >
> > We thank the reviewer for the prompt follow-up and the positive evaluation of our work!

---

### Official Review · Reviewer_JHNX · 2023-11-01

**Soundness:** 3 good
**Presentation:** 3 good
**Contribution:** 2 fair
**Rating:** 5
**Confidence:** 4

**Summary:**

This paper presents a way to do concept interventions on standard neural networks without the need for a Concept Bottleneck Model. This can be done by learning classifiers to predict the presence of a concept based on the hidden layer representation, and then optimizing to find representation that is as close as possible to original but changes the concept classifier prediction. They also propose to improve the effectiveness of interventions by finetuning the model to get better results under intervention.

**Strengths:**

Clearly written. Interesting perspective highlighting that intervention could be useful even on standard networks.

**Weaknesses:**

I don't really see any use cases where we would want to intervene on standard models instead of just creating a CBM.
- Intervention performance on models that weren't finetuned is quite poor, and still requires labeled concept data to learn the classifiers.
- The performance of models (even finetuned) is worse than CBM on most datasets, while both require additional training and the same kind of data with dense concept labels, most cases it would be better to just learn a CBM
- Intervening now requires solving an optimization problem making it more costly and harder to understand than original CBM interventions
- CBMs have many interpretability benefits in addition to intervene-ability, which we lose when using standard architecture, such as predictions being simple functions of interpretable concepts.

Lacking evaluation:
- I think improved performance on CheXpert is likely caused by the fact that the model can use information outside of concepts to make the prediction. This is similar to having residual as is done by Posthoc-CBM-h, and I think some comparison agaisnt that would be needed.
- Choice of datasets is a little odd, should use at least some of the datasets original CBM was trained on such as CUB

**Questions:**

Looks like each intervention requires running gradient descent to minimize eq. 1, what is the computational cost of this?
How did you intervene on multiple concepts at once?

---

> ### Author Response · Authors · 2023-11-17
> **Point-by-point Response to Reviewer JHNX (Part 1)**
>
> We thank the reviewer for the feedback! Below are our point-by-point responses to the concerns raised.
>
> > I don't really see any use cases where we would want to intervene on standard models instead of just creating a CBM.
>
> There exist use cases when training a CBM is not practical or possible. Firstly, concepts might be unknown to ML practitioners during model development due to the lack of domain knowledge, or it might be too expensive to annotate the training dataset. Secondly, with increased interest in and use of foundation models [[1]](https://doi.org/10.1038/s41586-023-05881-4), training dedicated task-specific models becomes less practical and feasible. Hence, there is a need for techniques that allow users to incorporate concept knowledge into representations and interact with predictive models without redesigning and retraining the entire model. This work explores this direction and proposes a practical fine-tuning approach to improve the intervenablity. At the same, the representations could still be used for other downstream tasks, as they remain unaffected by fine-tuning.
>
> > Intervention performance on models that weren't finetuned is quite poor, and still requires labeled concept data to learn the classifiers.
>
> It is true that in several datasets, the black-box models are less intervenable than CBMs. However, to the best of our knowledge, this work is the first to explore this research question, and even a somewhat negative finding in this regard is a valuable contribution to the literature, in our view. Finally, our fine-tuning approach mitigates this shortcoming of black-box models.
>
> Regarding the need for concept labels, we fully acknowledge that an annotated *validation set* is required (rather than an entire training set, as for CBMs). Additionally, our techniques are entirely compatible with the previous approaches for semiautomatic concept discovery, e.g. [[2](https://arxiv.org/abs/2205.15480), [3](https://arxiv.org/abs/2304.06129)]. Thus, in practice, the need for additional annotation can be alleviated by augmenting our methods with the ideas from the previous works. This research question is, however, beyond the scope of the current manuscript.
>
> > The performance of models (even finetuned) is worse than CBM on most datasets, while both require additional training and the same kind of data with dense concept labels, most cases it would be better to just learn a CBM.
>
> While it is true that fine-tuned models are less performant than CBMs on the synthetic data under the bottleneck and confounder mechanisms (and CUB, as observed in our additional experiments), there is a utility in fine-tuning models on the synthetic incomplete and chest X-ray datasets (and AwA, as observed in the updated experiments). The former scenarios are simplistic. In synthetic bottleneck and confounder, generative mechanisms directly or closely match the CBM. We expect CBMs to perform very well on those benchmarks. However, fine-tuned models are expected to perform better in more complex settings (synthetic incomplete and chest X-ray classification), where concept variables may only partially explain the relationship between $y$ and $\boldsymbol{x}$. After fine-tuning for those datasets, interventions allow injecting concept information into the representation without removing additional non-concept-based knowledge utilised by the black box.
>
> > Intervening now requires solving an optimization problem making it more costly and harder to understand than original CBM interventions.
>
> Intervening on black boxes indeed requires running the optimisation procedure; however, interventions do not require much time if the representations in the chosen layer are not too high-dimensional and can be performed on batches of instances. Moreover, since we use a multivariate probing function, interventions can be performed on multiple concept variables simultaneously without incurring additional computational complexity. In addition, on the user's side, there is no additional mental effort in understanding the interventions: the user can inspect the probe’s prediction of the concept variables and decide to intervene on a few of those, steering the model’s final prediction. Thus, interventions essentially work analogically to CBMs.
>
> > CBMs have many interpretability benefits in addition to intervene-ability, which we lose when using standard architecture, such as predictions being simple functions of interpretable concepts.
>
> We agree that CBMs have many advantages beyond interventions. Nevertheless, as mentioned above, there exist use cases when training a CBM from the beginning is not practical or impossible, especially when working with large neural networks whose representations are utilised in multiple downstream tasks. Thus, the proposed techniques can be an interesting alternative, covering a setting complementary to the typical CBM use case.

---

> > ### Author Response · Authors · 2023-11-17
> > **Point-by-point Response to Reviewer JHNX (Part 2)**
> >
> > > I think improved performance on CheXpert is likely caused by the fact that the model can use information outside of concepts to make the prediction. This is similar to having residual as is done by Posthoc-CBM-h, and I think some comparison agaisnt that would be needed.
> >
> > Similar to the synthetic incomplete, we expect black-box models to perform better than CBMs for the chest X-ray datasets since the former can extract information complementary to the concept variables. This phenomenon is one of the arguments for using our techniques over training a CBM *ante hoc*.
> >
> > To extend our empirical evaluation, we have included a comparison with post hoc CBMs [[2](https://arxiv.org/abs/2205.15480), [3](https://arxiv.org/abs/2304.06129)]. We have observed that while post hoc CBMs perform quite well on simpler datasets, for the synthetic incomplete and chest X-rays, interventions are either ineffective or harmful. We attribute this to potential leakage [[6]](https://proceedings.neurips.cc/paper_files/paper/2022/hash/944ecf65a46feb578a43abfd5cddd960-Abstract-Conference.html). While including a residual connection might improve the overall predictive performance, it would not affect the intervenability (or would even hurt it) since the residual prediction is just added to the final output of the post hoc CBM, as explained in [2]. Moreover, the parameters of the residual are optimised without retraining the concept-based module of the network (further suggesting that adding a residual connection should not affect the model’s behaviour under interventions). We will consider including a comparison with the hybrid post hoc CBMs in the appendix of the final manuscript in case of acceptance.
> >
> > > Choice of datasets is a little odd, should use at least some of the datasets original CBM was trained on such as CUB.
> >
> > We wanted to provide a well-rounded evaluation on the datasets with varying structures and difficulty. We included experiments on the (i) synthetic data for controlled exploration with known data-generating assumptions, (ii) AwA2, a simple natural image benchmark that is very similar to the CUB, and (iii) chest X-ray datasets to explore concept-based methods in the “wild”. We would like to note that these benchmarks have been explored in the previous related literature [[4](https://arxiv.org/abs/2212.07430), [5](https://arxiv.org/abs/2302.14460)]
> >
> > As suggested by the reviewer, we have also included experiments on the CUB dataset to corroborate our findings further. The results are described in the Appendix E.3 of the revised manuscript. Generally, the observed patterns, as expected, do not contradict our previous conclusions: we observe that fine-tuning for intervenability improves the effectiveness of interventions on black-box models and is superior to fine-tuning baselines.
> >
> > > Looks like each intervention requires running gradient descent to minimize eq. 1, what is the computational cost of this?
> >
> > Interventions require running a gradient descent subroutine to compute $\boldsymbol{z}’$. In practice, for speedup, the optimisation can be run on a batch of data points. The time until convergence varies depending on several parameters, e.g. $\lambda$ (higher $\lambda$s require more time until convergence) and $\varepsilon_I$ (see the relevant subroutine in Algorithm A.1). However, for all tested hyperparameter configurations, interventions could be easily performed online on a single GPU.
> >
> > > How did you intervene on multiple concepts at once?
> >
> > Since we utilise a *multivariate* probe function, interventions on multiple concept variables do not require additional consideration or tricks. Our methods (as presented in the manuscript) directly apply to multiple-variable interventions.
> >
> > ### References
> >
> > [1] Moor, M., Banerjee, O., Abad, Z. S. H., Krumholz, H. M., Leskovec, J., Topol, E. J., & Rajpurkar, P. (2023). Foundation models for generalist medical artificial intelligence. *Nature, 616*(7956), 259-265.
> >
> > [2] Yuksekgonul, M., Wang, M., & Zou, J. (2022). Post-hoc concept bottleneck models.
> > *arXiv:2205.15480*.
> >
> > [3] Oikarinen, T., Das, S., Nguyen, L. M., & Weng, T. W. (2023). Label-Free Concept Bottleneck Models. *arXiv:2304.06129*.
> >
> > [4] Chauhan, K., Tiwari, R., Freyberg, J., Shenoy, P., & Dvijotham, K. (2023). Interactive concept bottleneck models. In *Proceedings of the AAAI Conference on Artificial Intelligence* (Vol. 37, No. 5, pp. 5948-5955).
> >
> > [5] Marcinkevičs, R., Reis Wolfertstetter, P., Klimiene, U., Ozkan, E., Chin-Cheong, K., Paschke, A., ... & Vogt, J. E. (2023). Interpretable and Intervenable Ultrasonography-based Machine Learning Models for Pediatric Appendicitis. *arXiv:2302.14460*.
> >
> > [6] Havasi, M., Parbhoo, S., & Doshi-Velez, F. (2022). Addressing leakage in concept bottleneck models. *Advances in Neural Information Processing Systems, 35*, 23386-23397.

---

> > > ### Comment · Reviewer_JHNX · 2023-11-23
> > > **Response to rebuttal**
> > >
> > > Thanks for the response.
> > >
> > > I appreciate the changes to the submission by including additional datasets, results and baselines which I believe make the paper stronger.
> > >
> > > However this still has not addressed my main problems, i.e. when would this be useful. While you name potential cases, I think in almost all cases you would be better off with Post-hoc CBM[2] or LF-CBM[3] instead.
> > >
> > > While you have compared against a post-hoc CBM like model, I don't think this is a particularly fair comparison, as this is a strange implementation different from [2] and [3]. Most importantly, it is trained in a joint way, while [2][3] and trained sequentially, and joint training is known to lead to poor intervene-ability. For a better comparison you should instead train the CBM sequentially, or better yet use the actual methods of the papers. The most fair comparison would be Post-hoc CBM-h, which should be included as the residual can significantly improve performance.
> > >
> > > Last, my question regarding computational cost was not sufficiently addressed, and I would like to have some actual numbers on how long this takes.
> > >
> > > As a result, I retain my rating of below acceptance threshold.

---

> ### Author Response · Authors · 2023-11-21
> **Follow-up on the Rebuttal**
>
> Dear reviewer JHNX,
>
> Given that the end of the discussion period is approaching, we would like to ask if you have any further concerns or questions, particularly as a follow-up to our response? Thank you in advance!

---

> ### Author Response · Authors · 2023-11-22
> **Final Follow-up**
>
> Dear reviewer, we would like to double-check if you had a chance to assess the response and changes to the manuscript. Have these arguments and adjustments addressed your concerns?

---

> ### Author Response · Authors · 2023-11-23
> **Follow-up to the Response**
>
> Thank you for your follow-up!
>
> > I think in almost all cases you would be better off with Post-hoc CBM[2] or LF-CBM[3] instead.
>
> Our updated experiments show that training a CBM model post hoc does not result in the same intervention effectiveness as the proposed fine-tuning technique.
>
> > While you have compared against a post-hoc CBM like model, I don't think this is a particularly fair comparison, as this is a strange implementation different from [2] and [3]. Most importantly, it is trained in a joint way, while [2][3] and trained sequentially, and joint training is known to lead to poor intervene-ability. For a better comparison you should instead train the CBM sequentially, or better yet use the actual methods of the papers. The most fair comparison would be Post-hoc CBM-h, which should be included as the residual can significantly improve performance.
>
> We acknowledge that this implementation is different from the original paper. As in the rest of the baselines we included in our manuscript, we tried to adjust the architectures and training setup to be as fair as possible, i.e. including a non-linear layer after the bottleneck for enhanced expressivity or probing the original representations directly to the concept set.
>
> We would like to note that the current implementation is quite close to the label-free CBM, except for the joint vs. sequential training. We will investigate the use of sequential training in the camera-ready since the discussion period will end in half an hour.
>
> Regarding the hybrid approach, as mentioned in our previous response, including the sequentially trained residual connection would not affect intervenability since the residual prediction is added to the concept-based prediction at the very end. We would also like to stress that we were interested in investigating the effectiveness of interventions and not in the marginal gains in predictive performance (particularly since w/o interventions, most models perform quite comparably).
>
> > Last, my question regarding computational cost was not sufficiently addressed, and I would like to have some actual numbers on how long this takes.
>
> The computational cost varies across datasets and it is dependent on the amount of iterations performed for the optimization.
> The number of iterations required depends on several factors, e.g. the value of the parameter $\lambda$, the number of concept variables, the dimensionality of $z$, and the number of data points in the batch. Given all these considerations, providing a meaningful answer to this question is quite challenging. In our experience, interventions are considerably less costly than training or fine-tuning and can be performed online on a single GPU.
>
> For example, on the synthetic dataset, with the convergence parameter set to $10^{-6}$ (a very conservative value) and for a batch of 512 data points, the intervention procedure requires approx. 500 to 2500 steps (the number of steps varies depending on the number of given concepts and the value of $\lambda$), which amounts to 0.5 to 2 seconds (for the whole batch) for an untuned black-box model. We use smaller batches and more permissive convergence criteria when fine-tuning, allowing a considerable speedup. In addition, note that the run-time of the intervention procedure is not strictly dependent on the black-box model’s architecture (except for the dimensionality of the layer intervened on).

---

### Author Response · Authors · 2023-11-17
**To All Reviewers: Summary of Responses and Revisions**

We would like to thank you for thorough reviews and constructive feedback! Below, we summarise our responses to the main concerns and changes in the revised manuscript.

- **Setting and Relevance of Concept-based Interventions**: Some of you have questioned the utility of concept-based interventions on black boxes. We would like to emphasise that there exist settings where training a CBM ante hoc is impractical or impossible, for example, due to the lack of or limited domain knowledge at the model development time, scarcity of annotated data, or the use of foundation models [1]. We believe that our work provides findings that are helpful for such scenarios.

- **Strong Performance of CBMs**: Some of you were concerned about CBMs' stronger performance on some of the benchmarks. Firstly, along the lines of the point above, our techniques are complementary and are intended for settings where CBMs are impractical. Moreover, while CBMs did achieve strong results on *simpler* benchmarks, they were visibly less performant and intervenable than fine-tuned black boxes on the more realistic synthetic incomplete and chest X-ray datasets.

- **Additional Benchmarks**: Based on your feedback, we have incorporated experiments on the Caltech UCSD-Birds (CUB) dataset in Appendix E.3 of the revised manuscript. Our findings on this benchmark follow previous results, suggesting improved intervenability after applying the proposed fine-tuning strategy compared to other black boxes.

- **Updated Results for the AwA2 Experiment**: When incorporating the CUB dataset, we identified some logical errors in the code related to multiclass classification, which contaminated our original results on the AwA2 (the rest of the datasets have binary target variables and hence, for those, the original results are valid). We corrected those errors and updated all results for the AwA2 dataset. The updated findings do not change our general conclusions. However, we observe that interventions are now more effective on the model fine-tuned for intervenability than CBMs. We apologise for any inconvenience caused by this! During the discussion period, we double-checked the rest of the code and are firmly convinced it is error-free. We will release the updated and corrected code in case of acceptance.

- **Comparison with Post Hoc CBMs**: Based on some of your suggestions, we included an additional baseline: training a CBM post hoc in the spirit of [2] and [3]. This approach performs well on the simpler benchmarks, being slightly less intervenable than models fine-tuned for intervenability. However, interventions have an *adverse* effect on the performance of post hoc CBMs on more complex benchmarks, namely, synthetic incomplete and chest X-ray classification. Thus, these experiments further reinforce our point regarding the need for explicit inclusion of intervenability.

- **Relation to Conceptual Counterfactual Explanations**: Some of you have pointed out the similarity of our work to conceptual counterfactual explanations (CCE) considered in [4] and [5]. While some of the technical tricks we utilise are similar, the problem setting and the purpose of interventions and CCEs are different. The latter aim to produce plausible counterfactual explanations based on concepts. By contrast, we are concerned with injecting the *user-input* concept knowledge into the model’s representations. We have included references to this line of work in the revised paper and provided a detailed discussion in Appendix B.

- **Relationship between Intervenability and Feature Importance Measures**: Some of you have suggested relating the intervenability measure to feature importance. There indeed exist similarities, and the revised manuscript now discusses the relationship between intervenability and model reliance [6] in Appendix B.

For a detailed discussion, please refer to the individual point-by-point responses. Note that in the revised manuscript, changes are shown in blue. We look forward to hearing from you and will happily address any remaining feedback!

### References

[1]  Moor, M., Banerjee, O., Abad, Z. S. H., Krumholz, H. M., Leskovec, J., Topol, E. J., & Rajpurkar, P. (2023). Foundation models for generalist medical artificial intelligence. Nature.

[2] Yuksekgonul, M., Wang, M., & Zou, J. (2022). Post-hoc concept bottleneck models. ICLR.

[3] Oikarinen, T., Das, S., Nguyen, L. M., & Weng, T. W. (2023). Label-Free Concept Bottleneck Models. ICLR.

[4] Abid, A., Yuksekgonul, M. & Zou, J.. (2022). Meaningfully debugging model mistakes using conceptual counterfactual explanations. ICML.

[5] Kim, S., Oh, J., Lee, S., Yu, S., Do, J., & Taghavi, T. (2023). Grounding Counterfactual Explanation of Image Classifiers to Textual Concept Space. CVPR.

[6]  Fisher, A., Rudin, C., & Dominici, F. (2019). All Models are Wrong, but Many are Useful: Learning a Variable's Importance by Studying an Entire Class of Prediction Models Simultaneously. JMLR.

---

### Meta-Review · Area_Chair_zF8E · 2023-12-10

**Metareview:**

This paper proposes to intervene black-box model directly without the need to building a concept bottleneck model. This paper is generally well-written and some reviewers find the idea novel and interesting.

However, some reviewers also pointed out that it is not clear what is the real use of this idea, as intervention without interpretability is likely to lose trust and have potential problems like adversarial representations. Without interpretability, it is dangerous to proceed the black-box intervention despite showing promising improved performance. Another technical concern that is not resolved in the rebuttal period is that the proposed method still require concepts (although the authors claim that it is not necessarily needed) but both (i) training the probe, (ii) intervention, and (iii) fine-tuning all involve the information of concepts.

In the rebuttal period, the authors have provided additional experiment results following reviewers' request to compared with recent baselines, but there is still concerns on experiment setting, unfair comparisons and performance compared with interpretable CBM. The computational efficiency can be addressed better by providing a comparison table on different dataset and parameters.

Due to the above reasons, I recommend rejection for this paper. The authors are encouraged to incorporate the reviewers concerns to justify the motivation of this work.

**Justification For Why Not Higher Score:**

The motivation is not justified and there are some technical concerns.

**Justification For Why Not Lower Score:**

N/A

---

### Decision · Program_Chairs · 2024-01-16

Reject